# Vision Transformers Need Zoomer: Efficient ViT with Visual Intent-Guided Zoom Adapter

## Abstract

Vision Transformers (ViTs) have made significant strides recently, but vanilla ViT models struggle with complex scenes, particularly multi-label images and occluded objects. Humans can extract specific visual intents from complex images to guide effective classification. Inspired by human visual attention mechanisms that selectively focus on regions of interest while ignoring class-irrelevant areas, we propose ZoomViT, a novel approach that introduces visual intent-guided zoom adaptation for efficient vision transformers. ZoomViT is based on two key observations: (1) Humans and advanced models can intelligently ignore class-irrelevant areas and focus on semantically important regions through visual intent. (2) Standard ViTs can achieve superior classification accuracy when guided by adaptive zooming into regions that align with visual intent. Our approach introduces the Zoomer, a lightweight adapter with only 0.8M parameters that generates visual intent-guided score maps for image regions and dynamically adjusts patch sizes accordingly. This bio-inspired component simulates human-like visual attention by increasing patch density in regions deemed important by the visual intent guidance, while using larger patches for less critical areas. The visual intent-guided adaptation enhances both efficiency and accuracy, especially in complex images. Experiments show ZoomViT, based on the DeiT-S framework, achieves 83.8%(+4.0%) top-1 accuracy on ImageNet-1k, surpassing existing efficient state-of-the-art (SOTA) ViTs in accuracy and efficiency. The code will be publicly available.

## 1 Introduction

Vision Transformers (ViTs) treat images as sentences, allowing the use of NLP techniques for visual tasks Vaswani et al. (2017). The generality of the ViT architecture and its powerful feature extraction capabilities have achieved significant results in image classification, object detection, image generation, and other vision tasks Carion et al. (2020); Strudel et al. (2021); Peebles & Xie (2023); Zhao et al. (2021); Radford et al. (2021); He et al. (2022). The Vision Transformer divides the input image into uniform patches and embeds them into tokens through linear projection. This process is analogous to the tokenization of text in NLP, where words or subwords are converted into embeddings. By utilizing the self-attention mechanism, the Vision Transformer effectively captures the interdependencies among tokens in the input sequence. This modeling approach provides a robust foundation for downstream tasks Vaswani et al. (2017).

Natural images often contain objects with semantic information from multiple categories, creating potential ambiguity. However, inherent directional semantic cues—what we term visual intent—allow humans to effortlessly determine the primary category (e.g., in Figure 1, perceiving the image as a bird" rather than a roof"). While recent large-scale vision-language models exhibit similar capabilities, standard models often struggle with these relationships. Biologically, the human visual system solves this efficiency-accuracy trade-off not by processing the entire scene uniformly, but through foveal vision driven by top-down attention Wang et al. (2014); DiCarlo & Cox (2007). Humans dynamically allocate high-resolution processing power (the fovea) solely to the region of interest determined by their intent, while relegating the background to low-resolution peripheral vision Lecours et al. (1999); Bar (2003). Inspired by this biological mechanism, we hypothesize that

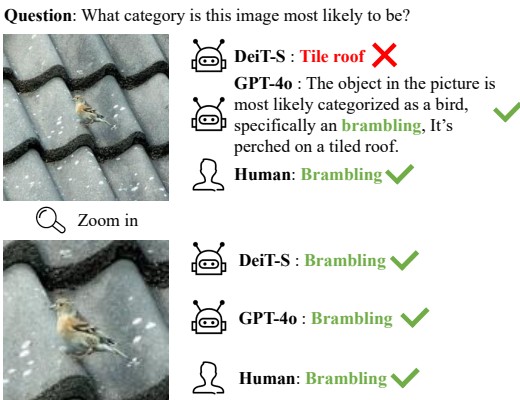

Figure 1: Comparison of image classification results between DeiT-S, GPT-4o and humans. The top image is the original image, and the bottom image is the local zoomed image guided by visual intent.

deep neural networks can achieve superior efficiency and robustness by mimicking this behavior: dynamically allocating finer patch granularity to intent-relevant regions while processing contextual areas with coarser resolution.

We identified two key phenomena related to visual intent in Vision Transformers. First, the vanilla ViT sets the default patch size to 16 pixels, dividing an image into $14 \times 14$ tokens. However, when an image contains multiple objects, the model's visual intent may be misaligned if the object of interest is not densely patched, leading to misclassification due to confusion with irrelevant objects. This suggests that ViT's visual intent can be redirected toward the correct target through appropriate patch sizing. To validate this hypothesis, we conducted a simple toy experiment using the DeiT-S model pre-trained on ImageNet Deng et al. (2009); Touvron et al. (2021a). In Figure 1, during the first stage, DeiT-S produced an incorrect classification result because its visual intent was drawn to the densely patched interfering category "tile roof" rather than the sparsely patched object of interest "Brambling". We used a simple method to verify this hypothesis. We zoomed the image to $1.5\times$ its original size and re-input it into the DeiT-S model. This resulted in denser patching for the "Brambling", effectively redirecting the model's visual intent toward the correct target, leading to accurate classification by the DeiT-S model. Therefore, we conclude that ViT models will perform better when their visual intent is properly guided through appropriate patch sizing.

The second phenomenon reveals that visual intent should focus on semantically meaningful pixels while ignoring redundant information. We discovered that challenging images often contain distracting pixels, such as the "tile roof" in Figure 1, which can mislead the model's visual intent. Conversely, some pixels are crucial for accurate classification and should be the primary focus of visual intent. Following this biological paradigmLecours et al. (1999); DiCarlo & Cox (2007); Bar (2003), we design a mechanism that enables ViT models to adaptively attend to semantically important pixels while suppressing irrelevant visual information. Combining this insight with our first observation, we propose using an adapter to identify potential class-decisive regions that align with proper visual intent. By zooming in on these regions, we can guide the ViT model's visual intent to focus on relevant information when classifying multi-label or occluded images. Notably, the computational cost of the ViT model increases quadratically with the number of input tokens. Therefore, zooming in on all regions is impractical. Thus, the adapter must precisely identify which regions deserve the model's visual intent to balance efficiency and accuracy.

Inspired by these phenomena, we introduce ZoomViT, a novel zoomable Vision Transformer that improves the prediction accuracy of complex image data. ZoomViT operates in two training stages. The first stage focuses on training the Zoomer. Zoomer is trained by distillation and aims to generate a heat map that captures visual intent for focus areas. The trained Zoomer assesses the class-decisive score of each patch in the input image. We establish a threshold $\alpha$ to identify regions requiring zooming. For regions scoring above $\alpha$, we apply a smaller patchify approach, while for unimportant regions, we use a larger patchify method. In the second stage, we re-rank patch tokens by their class-

decisive scores and adjust their representations with a zoom factor embedding before inputting them into the Transformer block. Our contributions are highlighted as follows:

- We introduce ZoomViT, which achieves a Top-1 accuracy of 83.8% on ImageNet-1k, surpassing the DeiT-S baseline by 4.0% with only a 0.8M increase in parameters. This performance establishes a new state-of-the-art (SOTA) in efficiency, combining high accuracy with minimal additional computational cost.

- We utilized Zoomer to assess the visual intent significance of tokens in ViT, using this as a foundation for effective pruning. This approach optimizes the model while maintaining its performance, enhancing the efficiency of Vision Transformer architectures.

- We identified the causes of ViT's classification errors when dealing with obscured targets or multi-labeled images. By employing ZoomViT, we have substantially mitigated these issues, demonstrating superior robustness and accuracy in challenging conditions.

## 2 RELATED WORK

The Transformer architecture Vaswani et al. (2017) has become essential for NLP tasks. With the Vision Transformer (ViT) Dosovitskiy et al. (2021), many vision tasks have adopted this architecture. However, the self-attention mechanism in ViT, while powerful, struggles with redundant image information and quadratic computational constraints, limiting its precision and efficiency with high-resolution, complex images. To address this, researchers have developed various ViT variants. Some methods focus on capturing hierarchical features Liu et al. (2021); Hatamizadeh et al. (2023). Other research focuses on efficient training paradigms Touvron et al. (2021a; 2022); Jiang et al. (2021). Recent studies Chen et al. (2023a); Xu et al. (2022) show that using varied patching methods can significantly enhance the accuracy of ViT for complex images.

In addition to developing high-performance network architectures and training paradigms, the way ViT processes images has also attracted attention. The difference in information density between the basic operational units (tokens) in image data units (patches) and text data units (words), the Transformer, originally invented in the NLP, has made certain compromises when processing image data. Specifically, Vanilla ViT employs a naive patchify method to split images. Recent studiesChen et al. (2023a); Xu et al. (2022) have demonstrated that using different patchify methods can effectively improve the accuracy of ViT in handling complex images.

## 3 METHODOLOGY

### 3.1 OVERVIEW

This section introduces ZoomViT, which addresses the limitation of vanilla ViT by treating each input patch with varying importance. As shown in Figure 2, ZoomViT's training involves two stages: training the Zoomer and training the Vision Transformer. In the first stage, class-decisive vectors guide the attention map generator to produce score maps for regions of interest. The class-decisive generator, known as Zoomer, is trained by optimizing the distillation loss. In the second stage, the trained Zoomer guides the patchification process for accurate image zooming and patch embedding. Positional embeddings are then added to the patch tokens. Token re-ranking uses the score map generated by Zoomer. After adding the zoom factor embedding, all tokens are fed into the Vision Transformer along with the $<CLS>$ tokens for training.

### 3.2 STAGE-1: ZOOMER TRAINING STRATEGY

Currently, various methods seek to identify key areas within an image. One intuitive approach uses pixel information content to indicate area importance Wang et al. (2023b;a). However, complex backgrounds often have high information entropy, rendering this method ineffective for accurate rankings. Other approaches Song et al. (2021); Huang et al. (2023) use learnable adapters to separate important from unimportant regions, but they lack dynamic partitioning and can produce redundant results. We innovatively apply Deep Taylor Decomposition to propagate relevancy scores in pretrained Vision Transformer layers Chefer et al. (2021), using class-decisive vectors to create class-

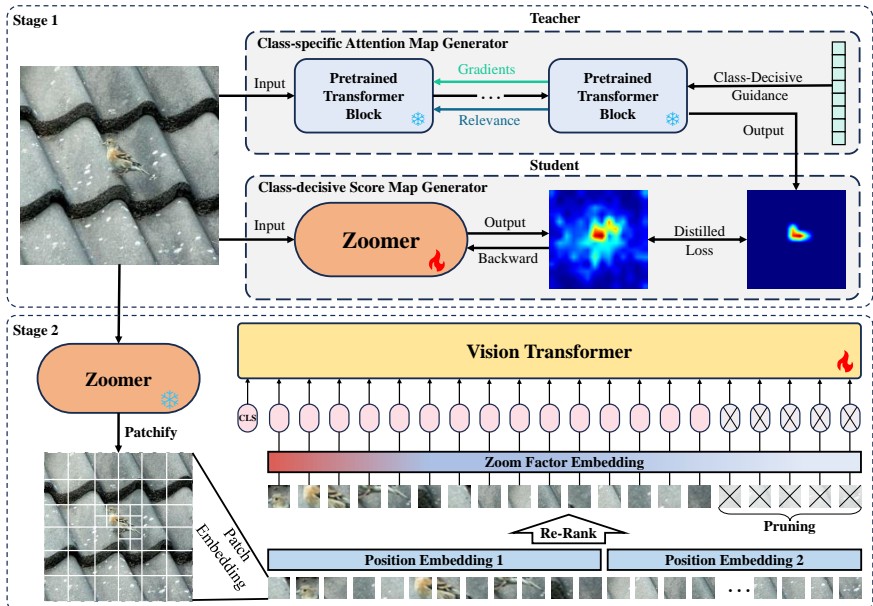

Figure 2: The overview of our framework. In stage 1, a Teacher-Student mechanism generates visual intent-driven class-specific attention maps. The teacher module uses a pretrained Transformer block to produce gradients and relevance that capture visual intent, guiding the student module's Zoomer to create class-decisive maps through distilled loss optimization. In stage 2, the Zoomer uses visual intent to intelligently patchify the input image, and the Vision Transformer processes these patches with positional and zoom factor embeddings to re-rank and produce the final classification result.

specific attention maps. We introduce a lightweight zoomer to identify regions of interest within an image, training it with distillation loss.

**Class-specific Attention Map Generator.** Assuming $C$ is the number of classes in the classification head of the pre-trained ViT, and the class of interest is $t \in 1... |C|$, we set the class-decisive guidance vector as $R^{(0)} = \delta_{it}$, where $\delta_{it}$ is the Kronecker delta function, defined as:

$$\delta_{it} = \begin{cases} 1 & \text{if } i = t \\ 0 & \text{if } i \neq t \end{cases} \tag{1}$$

Following Chefer et al. (2021), we denote $L^{(n)}(X, Y)$ as the operation of the $n$-th layer on the input feature map $X$ and weights $Y$. Relevance propagation is defined as follows:

$$R_j^{(n)} = \sum_i X_j \frac{\partial L_i^{(n)}(X, Y)}{\partial X_j} \frac{R_i^{(n-1)}}{L_i^{(n)}(X, Y)} \tag{2}$$

where index $j$ corresponds to elements in $R^{(n)}$, and index $i$ corresponds to elements in $R^{(n-1)}$.

Then, according to the relevance and gradient propagation process, we have:

$$\bar{A}^{(b)} = I + \mathbb{E}_h(\nabla A^{(b)} \odot R^{(n_b)})^+ \tag{3}$$

where $\odot$ is the Hadamard product, $A^{(b)}$ is the attention map of block $b$, and $\mathbb{E}_h$ is the average value across multiple heads in the dimension. The final output $C \in \mathbb{R}^{s \times s}$ is defined as the weighted attention relevance:

$$C = \overline{A}^{(1)} \cdot \overline{A}^{(2)} \cdot ... \cdot \overline{A}^{(B)} \tag{4}$$

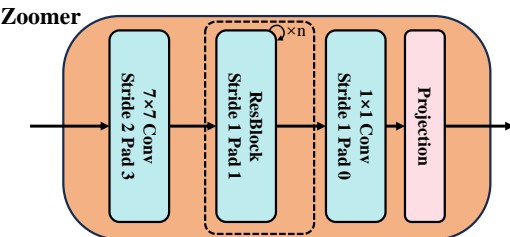

Figure 3: The architecture of the Zoomer module. consisting of an initial convolution layer, followed by a residual block repeated $n$ times, another $1 \times 1$ convolution layer, and a final projection layer.

In particular, we use the DeiT base pre-trained on ImageNet as a class-specific attention map generator, which serves as the teacher model. We freeze all gradients to prevent updates to the teacher model. We use the ground truth labels from the ImageNet training set, processed by Equation (1), to serve as the class-specific guidance vector. The class-specific attention map generator finally produces soft labels $P_t \in \mathbb{R}^{B \times S^2}$ to guide the training of the student model.

**Class-Decisive Generator.** We designed the zoomer in our class-decisive generator using a standard convolutional residual neural network. The model, defined as $\widehat{Y} = Z(x)$, where $\widehat{Y} \in \mathbb{R}^{B \times S^2}$, computes the class-decisive score map for each patch of the input image $x$. To reduce computational costs, we employed a lightweight network design, as shown in Figure 3. The output tensor $P_t$ is shaped to match the teacher model's output by stacking multiple residual blocks and incorporating a projection head. All convolutional layers, except the final $1 \times 1$ convolution, are followed by batch normalization Ioffe & Szegedy (2015) and activated by ReLU Nair & Hinton (2010). The number of residual modules $n$ is adjustable to balance parameters and performance.

**Loss Function.** Distillation minimizes the loss between the score map from the Zoomer and the score map from the class-specific attention map generator. The goal of distillation is:

$$
\begin{aligned}
\mathcal{L}_{global} =& \mathcal{L}_{mse}(\widehat{Y}, P) + w_1 D_{KL}(\widehat{Y} \parallel P_t) \\
& + w_2 \mathcal{L}_{Dice}(\widehat{Y}, P)
\end{aligned}
\tag{5}
$$

where, $\widehat{Y}$ represents the prediction from the Zoomer, and $P_t$ is the score map from the class-specific attention map generator. The loss function comprises three components, with $w_1$ and $w_2$ balancing the sub-losses. Similar to standard knowledge distillation methods Touvron et al. (2021a), we use Mean Squared Error (MSE) and Kullback-Leibler (KL) divergence to align the teacher and student model predictions. Additionally, to enhance the accuracy of overlapping regions between the Zoomer's score map and the teacher's score map, we incorporate Dice Loss to reduce fragmentation in the predicted heat map.

### 3.3 STAGE-2: ZOOMABLE VISION TRANSFORMER

After pre-training Zoomer, we use it to obtain the Class-Decisive score map from input images, which guides the zoom process. As shown in Figure 4, ZoomViT can control the patch size by using $\alpha \in [0, 1]$ to binarize the Class-Decisive score map and the Zoom factor $\eta \in \{0.5, 2.0\}$. Once we have the local zooming results, ZoomViT selectively prunes unimportant tokens to further reduce computational parameters. We employ different positional encodings for tokens of varying scales and sort them based on their scores on the Class-Decisive score map to prepare for Zoom factor embedding. For batch training, we pad sequences within the batch using $< pad >$ tokens.

**Zoom Factor Embedding.** To enable the model to discriminate image regions at different zoom levels, we need to add additional embeddings to the token sequences before inputting them to ViT. The simplest method is to add a fixed embedding to zoomed regions and another to non-zoomed regions. However, we discovered that the importance transition between zoomed and non-zoomed regions is quite gradual. In other words, zoomed regions merely indicate that certain positions of

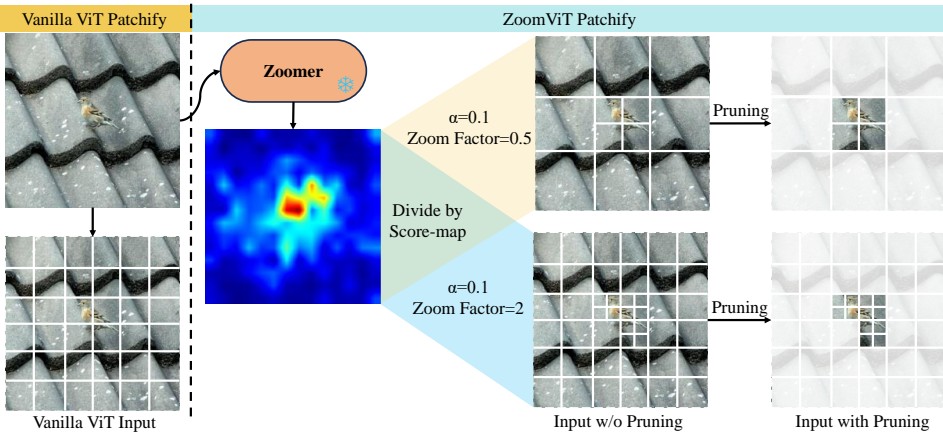

Figure 4: Patchifying process for ZoomViT. $\eta$ represents the zoom factor and $\alpha$ represents the threshold of the score map. ZoomViT accepts two types of inputs, w/o pruned inputs and pruned inputs.

the object are important, while other regions may also hold useful representations for the classifier. Thus, we developed a soft zoom factor embedding function, denoted as:

$$ZFE(pos) = 1 - \frac{1}{(1 + e^{\omega(N_k - pos)})} \tag{6}$$

where, $N_k$ represents the number of tokens in the zoomed region, and $\omega$ is a hyperparameter that controls the smoothness of the embedding.

**Inference Strategy.** When inference without pruning, ZoomViT utilizes patch tokens of different sizes, employing smaller patches in critical areas and larger ones in less significant regions. With $\eta = 0.5$, the input token count is lower than that of a standard ViT, leading to decreased computational demands. Conversely, with $\eta = 2$, the input token count surpasses that of a standard ViT, enhancing accuracy. If pruning is applied, ZoomViT ranks the tokens and removes the top $n$ least important ones to reduce computation.

## 4 EXPERIMENTS

### 4.1 IMPLEMENTATION DETAILS

In this section, we extensively validate the proposed ZoomViT on the ImageNet-1k Deng et al. (2009) classification dataset. ZoomViT is based on the DeiT-S Touvron et al. (2021a) model, with an embedding size of 384, 12 Transformer blocks, and 6-headed multi-head self-attention. We using a resolution of $224 \times 224$ for both training and testing. When training ViT, we use the same training parameters as DeiT to ensure fairness. During Zoomer training, we use the ImageNet pre-trained DeiT-B as a teacher model to generate class-specific attention maps. To accelerate training, we pre-extract class-specific attention maps from ImageNet-1k. We train the Zoomer model from scratch for 100 epochs with a batch size of 128. We use AdamW Loshchilov & Hutter (2019) for optimization, starting with an initial learning rate of 0.01.

During training, we randomly select a threshold $\alpha$ from the range $[0, 1]$ to ensure the model adapts effectively to various zoom scales. We trained two models using zoom factors of 0.5 and 2, respectively, with all other settings remaining the same as above. When testing FLOPs, since the zoomer dynamically generates the number of zoomed region tokens, we use a batch size of 1 to calculate the average FLOPs value on the ImageNet validation set. The model training in this work was performed on a workstation with 4 A100 GPUs. For testing accuracy, we used a batch size of 128 on a single A100 GPU. In the training phase of ViT, we follows the default configuration of DeiT[1].

---

[1]https://github.com/facebookresearch/deit

Table 1: Comparison of efficient ViT models based on Top-1 Accuracy, FLOPs, and parameter count. Here, $\eta$ represents the zoom factor and $\alpha$ indicates the threshold for the score map.

| Model | Top-1 Acc. (%) | FLOPs (G) | #Params (M) | Speed (img/s) |
|---|---|---|---|---|
| Deit-S | 79.8 | 4.6 | 22 | 5039 |
| DeiT III | 81.4 | 4.6 | 22 | 1891 |
| IA-RED$^2$ | 79.1 | 3.2 | 22 | 1360 |
| DynamicViT | 79.3 | 2.9 | 26.9 | 2062 |
| SPViT | 79.34 | 3.4 | 22 | - |
| PS-ViT | 82.3 | 8.8 | 21.3 | 464 |
| Evo-ViT | 79.4 | 3.0 | 22 | 1510 |
| ToMe | 78.89 | 2.9 | 22 | 6712 |
| EViT | 78.5 | 3 | 22 | 6807 |
| DiffRate | 79.58 | 2.9 | 22 | 6744 |
| ATS | 79.7 | 2.9 | 22 | - |
| LV-ViT | 83.3 | 6.6 | 26 | - |
| ToFu | 79.4 | 2.7 | 22 | 1552 |
| DeiT III-S 384 | 83.6 | 15.5 | 22 | 424 |
| CF-ViT | 80.8 | 4.0 | 22 | 2760 |
| **ZoomViT** | | | | |
| $\eta$=0.5, $\alpha$=0.03 | 81.5 | 2.3 | 22.7 | 6738 |
| $\eta$=2, $\alpha$=0.1 | 82.5 | 6.3 | 22.7 | 3721 |
| $\eta$=2, $\alpha$=0.01, Pruning | **83.8** | 6.3 | 22.7 | 3717 |

## 4.2 EXPERIMENTAL RESULTS

**Comparison with efficient ViT models.** To illustrate the capability of our ZoomViT in balancing model accuracy and complexity, we compare it with recent efficient Vision Transformer (ViT) models. When the zoom factor is set to $\eta = 0.5$, ZoomViT functions as a token pruning method. Table 1 lists token pruning methods with similar parameter counts, such as IA-RED$^2$ Pan et al. (2021), DynamicViT Rao et al. (2021), SPViT Kong et al. (2022), PS-ViT Tang et al. (2022), EVO-ViT Xu et al. (2022), ToMe Bolya et al. (2023), ToFuKim et al. (2024), EViT Liang et al. (2022), DiffRate Chen et al. (2023b), ATS Fayyaz et al. (2022), and data-efficient methods, including DeiT Touvron et al. (2021a), DeiT III Touvron et al. (2022), LV-ViT Jiang et al. (2021). We present the top-1 accuracy, FLOPs, and model parameters. Comparative results with recent efficient transformer-based models indicate that ZoomViT significantly outperforms previous methods. Specifically, with a zoom factor of $\eta = 0.5$, ZoomViT's FLOPs are significantly lower than the baseline. When $\alpha$ is set to 0.01, ZoomViT achieves 81.5% (+1.7%) accuracy while generating only 2.3G (-50%) FLOPs. ZoomViT also balances efficiency and performance. When $\eta = 2$, $\alpha = 0.01$, we turn on pruning to control the number of inference tokens to be consistent with $\eta = 2$, $\alpha = 0.1$, ZoomViT achieves 83.8% (+4.0%) accuracy. We discovered a counterintuitive result: the accuracy after pruning was higher than before pruning, precisely demonstrating Zoomer's effectiveness in eliminating negative tokens. Specifically, Zoomer achieves positive gains by pruning misleading tokens. Compared to the baseline, ZoomViT has only a slight increase in parameters due to the additional zoomer. For methods that also increase the number of parameters, such as DynamicViT and LV-ViT, the additional parameters introduced by the zoomer are acceptable.

**Comparison with SOTAs.** To demonstrate the competitiveness of our ZoomViT, Figure 5 illustrates the balance between accuracy and FLOPs by varying $\alpha$ across different zoom levels. We compare ZoomViT with mainstream state-of-the-art methods, such as DeiT Touvron et al. (2021a; 2022), Swin Transformer Liu et al. (2021), CaiT Touvron et al. (2021b), T2T-ViT Yuan et al. (2021), CrossViT Chen et al. (2021), PVT Wang et al. (2021a), ViG Han et al. (2022), and EfficientFormer Li et al. (2022). As shown, when a zoom factor of $\eta = 0.5$, ZoomViT demonstrates superior speed compared to efficient methods like Swin-Tiny, DeiT-S, and PVT-S. With a zoom factor of $\eta = 2$, ZoomViT achieves better performance while maintaining a lower computational cost. Our designed zoomer guides the model's decision-making towards greater accuracy. The **Appendix** shows comparisons with more datasets and more baselines.

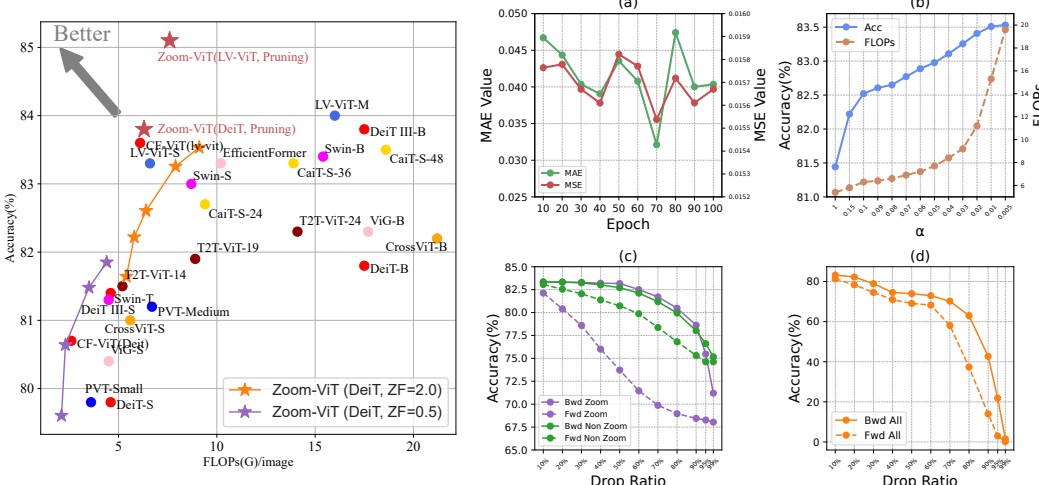

Figure 5: Comparison with SOTA ViT models.

Figure 6: Ablation studies on (a) training epoch, (b) accuracy vs FLOPs, and (c)(d) token pruning.

Table 2: Comparison of performance across varying numbers of residual blocks $n$.

| Ablation | MSE | MAE | #Params (M) |
|---|---|---|---|
| $n$=2 | 0.0161 | 0.0468 | 0.3 |
| $n$=4 | 0.0155 | 0.0321 | 0.8 |
| $n$=6 | 0.0112 | 0.0249 | 2.9 |

Table 3: Comparison of performance across various $\alpha$ values during training.

| Ablation | Top-1 Acc. (%) |
|---|---|
| Fixed $\alpha = 0.03$ | 83.1 |
| Fixed $\alpha = 0.1$ | 82.52 |
| Random $\alpha$ | 83.348 |

Table 4: Comparison of performance across various zoom factor embedding techniques.

| Ablation | Top-1 Acc. (%) |
|---|---|
| Fixed ZFE | 83.23 |
| w/o ZFE | 83.07 |
| Our ZFE | 83.348 |

Table 5: Comparing the performance of various pruning methods.

| Ablation | Top-1 Acc. (%) |
|---|---|
| Baseline | 81.4 |
| w/o Pruning (Avg) | 82.74 |
| Pruning (Avg) | 82.97 |

## 5 ABLATION STUDY

To validate the effectiveness of each design in ZoomViT, we performed ablation studies on the zoomer, hyperparameters, zoom factor embedding, and token re-ranking and pruning. In this section, we illustrate the necessity of each design element by removing or replacing it. The training settings for the ablation studies were identical to those used in DeiT Touvron et al. (2022). Please refer to the **Appendix** for more complete ablation experiments and visualisation results.

### 5.1 ZOOMER

**Influence of training epoch.** To ensure that Zoomer accurately identifies the class decisive for classification, we use the mean absolute error (MAE) and the mean squared error (MSE) to measure the difference between Zoomer's class-decisive score map and the soft labels from the teacher model. Figure 6 (a) shows the changes in MSE and MAE throughout the training epochs. The figure indicates that the model reaches optimal performance at epoch 70. Thus, unless stated otherwise, we use the checkpoints from Zoomer trained for 70 epochs as the default in the following sections.

**Influence of architectural.** Designing additional adapters increases the computational load on ZoomViT. Table 2 presents the performance and parameter count of ZoomViT with varying numbers of residual blocks. To optimize efficiency, we select the configuration with 4 residual blocks as the default.

**Influence of $\alpha$.** Figure 6 (b) illustrates the relationship between accuracy and FLOPs with a zoom factor of $\eta = 2$. Various threshold values of $\alpha$ distinguish between zoomed and non-zoomed regions. As depicted in Figure 6 (b), a smaller $\alpha$ classifies more patches as requiring zooming, which increases accuracy but also raises FLOPs consumption. In this study, unless stated otherwise, we set $\alpha$ to 0.03 to balance accuracy and efficiency.

We conducted ablation studies to evaluate the impact of using a random $\alpha$ strategy during training. To maintain controlled conditions, we consistently applied a zoom factor of $\eta = 2$ and varied only the $\alpha$ parameter during training. For validation, we fixed $\alpha$ at 0.03. Table 3 presents the results demonstrating the effectiveness of the random $\alpha$ allocation strategy during training. Detailed ablation studies for $\alpha$ and $\eta$ are provided in the **Appendix**.

### 5.2 ZOOM FACTOR EMBEDDING

Table 4 presents the ablation study on various zoom factor embedding techniques. We tested the absence of zoom factor embedding, fixed zoom factor embedding, and our proposed zoom factor embedding. Our results indicate that our proposed method improves accuracy by 0.28% over no zoom factor embedding and by 0.12% over fixed zoom factor embedding.

### 5.3 TOKEN RE-RANKING AND PRUNING

To show that Zoomer can generate accurate rankings, we conducted an experiment where tokens were pruned based on their scores in both forward and backward directions, as illustrated in Figure 6 (c) and (d). In this context, pruning in the forward direction means removing tokens starting from the highest-ranked (most important) to the lowest-ranked based on their scores, while backward pruning removes tokens from the lowest-ranked to the highest-ranked. We observed that pruning tokens in the forward direction led to a larger drop in accuracy compared to pruning in the backward direction. This indicates that the score map accurately ranks tokens by their importance. Additionally, pruning tokens in the zoomed regions caused a larger accuracy decrease compared to non-zoomed regions, proving that Zoomer effectively identifies class-decisive regions. Similarly, as shown in Figure 6 (d), the same phenomenon was observed when all tokens were concatenated and then pruned.

We tested the performance differences between using pruned and non-pruned patches as inputs. Table 5 presents the average accuracy for $\alpha$ values ranging from 0.05 to 1 in 0.01 intervals. Our method achieves an accuracy of 82.97%. The increase in scores after pruning is attributed to ZoomViT eliminating irrelevant tokens, thereby allowing ViT to more effectively predict categories.

### 5.4 EFFECTIVENESS OF ZOOMER

In the **Appendix** Figure 10, we visualized the class-decisive map predicted by Zoomer. We have selected several images that ZoomViT can correctly classify but DeiT cannot. Obviously, these images often contain obscured targets or multi-label confusion category information. Our ZoomViT successfully relies on the zoomer's guidance to perform local zooming, thereby solving problems that DeiT cannot. The **Appendix** provides additional experiments and visualizations related to Zoomer.

## 6 CONCLUSION

In this study, we introduce ZoomViT, an innovative zoomable Vision Transformer. ZoomViT employs a Zoomer to create visual intent-guided score maps, effectively identifying semantically important regions and redirecting the model's attention toward class-decisive areas. Essential designs like token re-ranking and adjustable zoom factors enable ZoomViT to balance efficiency and performance. Experimental results show that ZoomViT excels in managing images with multiple labels and obscured targets, significantly outperforming the DeiT model. This discovery opens new avenues for efficient Vision Transformers. Furthermore, applying the zoom concept to advanced visual tasks such as concealed object detection, fine-grained image classification, and object tracking is part of our future work.

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

# A  APPENDIX

# B  MORE RELATED WORK

Recent research focuses on reducing redundant information in model's inference decision-making process. Some studies suggest using patches of different sizes in ViT to capture regions with varying information densities Chen et al. (2023a); Song et al. (2021). Other studies rank token importance in intermediate layers to discard less important tokens, thereby narrowing the focus of the model Rao et al. (2021); Pan et al. (2021); Liang et al. (2022); Xu et al. (2022). Collectively, these studies demonstrate that token importance varies within Vision Transformers, and different patching methods can yield different results for the same image.

## B.1  VISION TRANSFORMER

The Transformer architecture Vaswani et al. (2017) has become essential for NLP tasks. With the Vision Transformer (ViT) Dosovitskiy et al. (2021), many vision tasks have adopted this architecture. However, the self-attention mechanism in ViT, while powerful, struggles with redundant image information and quadratic computational constraints, limiting its precision and efficiency with high-resolution, complex images. To address this, researchers have developed various ViT variants. Some methods focus on capturing hierarchical features Liu et al. (2021); Hatamizadeh et al. (2023). Other research focuses on efficient training paradigms Touvron et al. (2021a; 2022); Jiang et al. (2021). Recent studies Chen et al. (2023a); Xu et al. (2022) show that using varied patching methods can significantly enhance the accuracy of ViT for complex images. This study also introduces a general adapter to adapt various ViT frameworks by modifying only the input tokens.

In addition to developing high-performance network architectures and training paradigms, the way ViT processes images has also attracted attention. The difference in information density between the basic operational units (tokens) in image data units (patches) and text data units (words), the Transformer, originally invented in the NLP, has made certain compromises when processing image data. Specifically, Vanilla ViT employs a naive patchify method to split images. Recent studies have demonstrated that using different patchify methods can effectively improve the accuracy of ViT in handling complex images.

## B.2  EFFICIENT ViT

The Swin Transformer Liu et al. (2021) reduces computational complexity using shifted windows, facilitating local self-attention while capturing global information. DW-ViT Ren et al. (2022) uses a dynamic window strategy to handle multi-scale information, whereas DVT Wang et al. (2021b) employs three Transformers for patch size allocation, which increases storage overhead. Methods such as DynamicViT Rao et al. (2021) and DGE Song et al. (2021) focus on token pruning to improve efficiency. CFViT Chen et al. (2023a) and Evo-ViT Xu et al. (2022) enhance image recognition by integrating dynamic patching with computational budget allocation.

These approaches share a common goal: filtering tokens during inference based on their importance. However, these methods encounter issues. Although they aim to reduce computational costs, their performance often barely surpasses the baseline. Moreover, during inference, the number of input tokens remains static, regardless of image complexity. In contrast, our ZoomViT provides various zoom levels, balancing efficiency and performance, similar to a camera's "zoom out" and "zoom in" functions.

## C  PRELIMINARIES

ViT Dosovitskiy et al. (2021) was the first to propose applying the Transformer Vaswani et al. (2017) model's encoder to computer vision. ViT slices images into patches, then reshapes them into a one-dimensional sequence and linearly projects them into a hidden space embedding. An additional $<CLS>$ token is concatenated with the input tokens to capture global information. All tokens are augmented with learnable positional encoding to assist model training. Therefore, an input token in ViT can be represented as:

$$X = [<CLS>; f(x_{0,0}^0); f(x_{1,0}^1); ...; f(x_{i,j}^N)] + E_{pos} \tag{7}$$

Where, $f(x_{i,j}^N) \in \mathbb{R}^D$ is the D-dimensional token of the $(i, j)$-th patch of the original image, $N$ is the number of input patch tokens, $f(\cdot)$ is the linear projection module, and $E_{pos}$ is the positional embedding.

The ViT model comprises a series of stacked Transformer Encoder blocks, each comprising a Self-Attention (SA) module and a feed-forward network (FFN). Given an input token sequence $X$, the $K$-th layer encoder can be represented as:

$$Y_k = X_{k-1} + MSA(LayerNorm(x_{k-1})) \tag{8}$$

$$X_k = Y_k + FFN(LayerNorm(Y_k)) \tag{9}$$

Where, the SA module maps the input sequence into vectors $Q, K, V \in \mathbb{R}^{(1+N) \times D}$ using three learnable linear projections. The weighted sum of all values in the sequence is then computed as follows:

$$SA(Q, K, V) = Softmax(\frac{Q \cdot K^T}{\sqrt{D}})V \tag{10}$$

### C.1  COMPUTATIONAL COMPLEXITY

The efficiency and accuracy of ViT are closely related to the number of input patches. Fewer patches lead to higher computational efficiency, while more patches lead to higher accuracy Xu et al. (2022); Rao et al. (2021); Touvron et al. (2022); Tang et al. (2022). Given an image segmented into $N$ patches, the computational complexity of SA and FFN in each transformer block is:

$$O(SA) = 3ND^2 + 2N^2D \tag{11}$$

$$O(FFN) = 8ND^2 \tag{12}$$

The efficiency of SA is inversely proportional to the square of the number of $N$, while FFN is linearly related to the number of $N$. Therefore, the efficiency of ViT at both standard resolutions ($224 \times 224$ in vanilla ViT) as well as at higher resolutions, which is one of the main focuses of our work.

## D  RESULTS ON MORE DATASETS

To better reveal the fundamental principles behind ZoomViT's zoom mechanism for performance improvement, we conducted further experiments on the ImageNet-A Hendrycks et al. (2021) dataset. ImageNet-A provides numerous Natural Adversarial Examples that are visually clear to humans but significantly reduce the accuracy of mainstream models. The ImageNet-A dataset contains complex perturbations that occur in real-world scenarios (occlusion, rare viewpoints, background interference). We tested ZoomViT's performance on ImageNet-A, as shown in Table 6. ZoomViT achieved significant improvements on ImageNet-A. To specifically demonstrate the working principle of ZoomViT's zoom mechanism, we visualized the intermediate process results of ImageNet-A

Table 6: Comparison of classification accuracy on ImageNet-A dataset.

| Method | ImageNet-A (Acc%) |
|---|---|
| AlexNet | 1.77 |
| VGG16 | 2.63 |
| DenseNet121 | 2.16 |
| ResNet-50 | 2.17 |
| ResNet-152 | 6.05 |
| DeiT-tiny | 7.25 |
| DeiT-small | 19.10 |
| DeiT-small+TTA | 21.10 |
| **ZoomViT** | **23.11** |

validation set images during ZoomViT inference in Figure 13. We found that this performance improvement primarily stems from ZoomViT's adaptive zoom mechanism effectively addressing three main challenges in ImageNet-A:

- **Occlusion Robustness**: When target objects are partially occluded, ZoomViT's local adaptive zoom can dynamically adjust the feature weights of occluded and visible regions, enabling the model to focus more on effective visual information.

- **Viewpoint Adaptability**: When facing samples with rare viewpoints, ZoomViT dynamically adjusts the importance weights of patches through patch rerank and zoom factor embedding, thereby better recognizing target objects under different viewpoints.

- **Background Interference Suppression**: Interference information in complex backgrounds is a significant factor causing model misjudgment. ZoomViT's zoom mechanism can adaptively enhance the expression intensity of foreground target features while suppressing the influence of background patches through token pruning.

# E    RESULTS ON DOWNSTREAM TASK

To verify the effectiveness of ZoomViT design in downstream task transfer, we selected YOLOSFang et al. (2021), which also adopts the pure DeiT-S architecture, as the comparison baseline for object detection. Specifically, we replaced the feature extractor of YOLOS with pre-trained ZoomViT and conducted 150 epochs of training on the COCO dataset, with other implementation details remaining consistent with the original YOLOS. As shown in Table 7, the experimental results demonstrate that when ZoomViT is transferred to position-sensitive downstream tasks such as object detection, its local zoom mechanism significantly improves model performance, with mean average precision increasing by 0.4%.

Table 7: Comparative transferability of ZoomViT in downstream tasks (object detection).

| Method | mAP |
|---|---|
| YOLOS (DeiT-S) | 36.1 |
| YOLOS (ZoomViT-S) | 36.5 |

# F    MORE ABLATION STUDIES

## F.1    COMPARISON OF DIFFERENT SIZE MODELS

To verify the scalability of the ZoomViT paradigm across models with different parameter counts, we replaced the ZoomViT backbone with DeiT-Tiny and DeiT-Base for testing. As shown in Table 8, applying the ZoomViT paradigm to both smaller and larger parameter models achieved significant performance improvements. This result demonstrates that ZoomViT's effectiveness does not depend on feature extractors with specific parameter counts, but rather stems from the advantages of the local zoom paradigm itself.

Table 8: Performance comparison of ZoomViT using different sizes of backbone as feature extractors.

| Backbone | Baseline | ZoomViT |
|---|---|---|
| DeiT-tiny | 72.20 | 79.14 |
| DeiT-small | 79.80 | 83.80 |
| DeiT-base | 82.89 | 84.90 |

## F.2 Ablation of $\alpha$ and $\eta$

Different threshold $\alpha$ and zoom factor $\eta$ affect the resolution and number of patches, so we compared model accuracy under different $\alpha$ and $\eta$ parameter combinations to better demonstrate the trade-off relationship between performance and efficiency. As shown in Table 9, under the same $\eta$ value, the smaller the threshold $\alpha$, the finer the patch division, and the more fine-grained local image information the model can obtain. $\eta$ controls the base patch size, and when $\eta = 2$, images of the same resolution are divided into more patches, thus achieving higher performance.

Table 9: Ablation of $\alpha$ and $\eta$

| $\eta \backslash \alpha$ | 1 | 0.15 | 0.05 | 0.01 | 0.005 |
|---|---|---|---|---|---|
| 2 | 81.44 | 82.22 | 82.968 | 83.80 | 83.88 |
| 0.5 | 71.30 | 74.29 | 78.04 | 81.50 | 81.85 |

## F.3 Computational Efficiency of Zoomer

We compared the impact of different Zoomer architectures on computational efficiency. As shown in Table 10, the model's performance gradually improved with the incremental addition of Zoomer layers. Once the model reached a certain number of layers, the performance stabilized.

## F.4 Comparison with Other Key Area Identification Methods

We tested the inference speed and performance of Zoomer against the key region extraction methods in DQVAEHuang et al. (2023) and HPMWang et al. (2023a) with a batch size of 8. Each input image was processed in a loop 100 times to calculate the average inference speed. As shown in Table 11, Zoomer demonstrates significantly shorter runtime per image compared to other methods while achieving higher accuracy. We attribute this to the fact that methods like DQVAE and HPM focus on identifying complex regions within an image, rather than emphasizing key areas. Background or insignificant regions can also have complex textures, which serve as distractions for the model.

## F.5 Why ZoomViT Works in Complex Scenes?

Large-scale natural image datasets, such as ImageNet, are typically annotated with single-label tags. This labeling approach was initially deemed reasonable; however, studies, such as Shankar et al. (2020), have revealed that approximately 20% of images in ImageNet possess more than one valid label. A substantial body of work, including Yun et al. (2021); Tsipras et al. (2020), has focused on addressing the impact of multi-label images on classifiers. Proposed solutions generally fall into two categories: one emphasizes training models to focus on the correct category within multi-label images, while the other involves augmenting validation sets with multi-label annotations. ZoomViT, by emulating camera zoom functionality, automatically focuses on critical regions within an image, and putting the unimportant areas "out of the frame". This approach mitigates the confusion introduced by multi-label images to a certain extent and demonstrates significant effectiveness in classifying occluded objects. In natural images, occluded objects are often obscured by backgrounds or other prominent entities, causing traditional classification models to fail due to their lack of targeted attention to these regions. In contrast, ZoomViT's zoom mechanism first identifies potential targets within a global view and then progressively zooms into local regions to uncover hidden detailed features.

Table 10: Comparison of computational efficiency across varying numbers of residual blocks $n$ in Zoomer.

| Ablation | #Params (M) | FLOPs (G) | ImageNet (Acc%) |
|---|---|---|---|
| $n$=2 | 0.31 | 0.58 | 83.348 |
| $n$=4 | 0.83 | 1 | 83.42 |
| $n$=6 | 2.93 | 1.4 | 83.45 |

Table 11: Comparison of performance across different key area identification methods.

| Ablation | Inf Time (s) | ImageNet (Acc%) |
|---|---|---|
| Zoomer | 0.00102 | 83.348 |
| DQVAE | 0.02341 | 79.3 |
| HPM | 0.01079 | 81.42 |

### F.6 IMPORTANCE OF 2-STAGE TRAINING

ZoomViT adopts a two-stage training strategy, consisting of Zoomer training and ZoomViT training. The first stage training enables the Zoomer to acquire the ability to discriminate the importance levels of image patches. Thanks to the extremely lightweight Zoomer structure design, this stage requires only 2 GPU hours to complete training. The second stage uses the pre-trained and frozen-weight Zoomer to train the ViT model. Since all modifications in ZoomViT are performed before input to the Transformer, theoretically the Zoomer can be utilized in a test-time augmentation (TTA) manner on the original DeiT to achieve effects similar to ZoomViT. To verify the necessity of second-stage fine-tuning, we demonstrate the performance of original DeiT in TTA mode on ImageNet-A in Table 13. Results show that ZoomViT trained through the second stage can better adapt to the dynamic patch resolution generated by local zoom. Therefore, although two-stage training moderately increases training costs, this additional overhead is completely acceptable relative to the performance improvement.

Table 12: Comparison of total training time between DeiT-S and ZoomViT.

| Method | DeiT-S | ZoomViT |
|---|---|---|
| Total Training Time (GPU-h) | $\sim$212 | $\sim$214 |

### F.7 EXPANSION OF TOY EXPERIMENTS

To better illustrate our research motivation, we conducted an extended validation of the toy experiment mentioned in the introduction. Specifically, we first uniformly scaled input images to 384×384 resolution, then used overlapping segmentation to divide images into 3×3 patches totaling 9 patches of 224×224 each, and fed these patches separately into the DeiT-S model for inference. In terms of evaluation criteria, we considered the prediction for the entire image correct as long as DeiT-S could correctly predict any one of the 9 patches. Essentially, this operation is equivalent to locally zooming 9 overlapping regions in the original image. The experimental results shown in Table 14demonstrate that images processed with local zoom achieved significant performance improvements on DeiT-S, but correspondingly incurred 9 times the computational overhead. This complete toy experiment strongly validates our core hypothesis: locally zooming natural images can effectively improve model prediction performance.

## G ADAPTATION TO RESOLUTION-SENSITIVE ARCHITECTURES

While ZoomViT demonstrates strong performance with standard ViT architectures, advanced models such as LV-ViT and Swin Transformer impose stricter constraints on input token structures. These architectures rely fundamentally on fixed token sequences and regular spatial grids, which appear incompatible with ZoomViT's dynamic resolution paradigm. In this section, we present a principled adaptation strategy that preserves the intention-aware capabilities of ZoomViT while satisfying the structural requirements of resolution-sensitive models.

Table 13: A performance comparison of implementing and not implementing the second stage of training on ImageNet-A.

| Method | ImageNet-A (Acc%) |
|--------|-------------------|
| DeiT+TTA | 21.10 |
| ZoomViT | 23.11 |

Table 14: The complete toy experiment mentioned in the introduction was conducted on the ImageNet dataset.

| Method | ImageNet (Acc%) |
|--------|-----------------|
| DeiT-S | 79.80 |
| DeiT-S 9-Crop | 88.61 |

The core challenge stems from architectural dependencies. LV-ViT's Token Labeling mechanism requires a pre-defined token sequence with fixed cardinality to generate auxiliary supervision signals. Similarly, Swin Transformer's Shifted Window attention operates on strict $M \times M$ grid partitions, where dynamic or non-uniform patch distributions would necessitate complex boundary handling and fundamentally disrupt the computational efficiency afforded by its regular windowing scheme. Direct application of ZoomViT's variable-resolution patches would therefore compromise both the training paradigm of LV-ViT and the parallel computation efficiency of Swin Transformer.

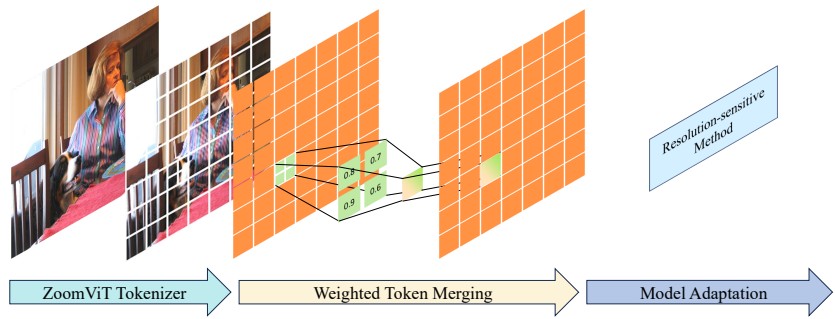

Figure 7: Workflow of intention-guided multi-resolution patch embedding and token fusion for LV-ViT and Swin Transformer.

To address this incompatibility, we introduce a score-weighted token fusion strategy inspired by recent advances in token merging methodsKim et al. (2024); Bolya et al. (2023). As shown in Figure 7The adaptation process proceeds in two stages. First, we leverage the pre-trained Zoomer to generate the visual intent-guided score map and perform multi-resolution patch embedding following the standard ZoomViT procedure. At this stage, regions identified as semantically important receive quadruple the patch density compared to baseline models. Second, before feeding the token sequence into the downstream Transformer blocks, we apply a spatial aggregation operation. Specifically, we group tokens corresponding to each $2 \times 2$ spatial region in the zoomed areas and fuse them into a single token through weighted averaging, where the fusion weights are derived from the normalized scores in the Zoomer's output map. This score-weighted fusion ensures that the final token inherits richer representations from patches covering more discriminative visual content.

Mathematically, given a set of four tokens $\{\mathbf{t}_1, \mathbf{t}_2, \mathbf{t}_3, \mathbf{t}_4\}$ corresponding to a $2 \times 2$ patch group with respective scores $\{s_1, s_2, s_3, s_4\}$ from the score map, the fused token $\mathbf{t}_{\text{fused}}$ is computed as:

$$\mathbf{t}_{\text{fused}} = \sum_{i=1}^{4} \frac{s_i}{\sum_{j=1}^{4} s_j} \mathbf{t}_i \tag{13}$$

This formulation ensures that patches within important regions contribute more substantially to the fused representation, thereby preserving the intention-guided focus of ZoomViT while producing a token sequence with cardinality identical to that expected by the baseline architecture.

The resulting adaptation strategy offers several advantages. It maintains the visual intention guidance mechanism that underlies ZoomViT's effectiveness on complex images, as the score-weighted fusion explicitly prioritizes discriminative visual information during token aggregation. Simultaneously, it produces token sequences that conform precisely to the structural requirements of target architectures, enabling seamless integration with existing training frameworks and preserving the computational optimizations inherent to models like Swin Transformer. This approach represents a practical balance between preserving the core innovations of ZoomViT and ensuring compatibility with diverse architectural paradigms in the Vision Transformer landscape.

## H  VISUALISATION

In this section, we present additional classification samples where ZoomViT successfully classifies images, whereas DeiT-S fails. We display the score maps of samples predicted by Zoomer. Green text indicates the correct class name, while red text indicates the incorrect class name. As shown in Figure 12. Zoomer effectively identifies hidden objects in the images, guiding ZoomViT to perform local zoom.

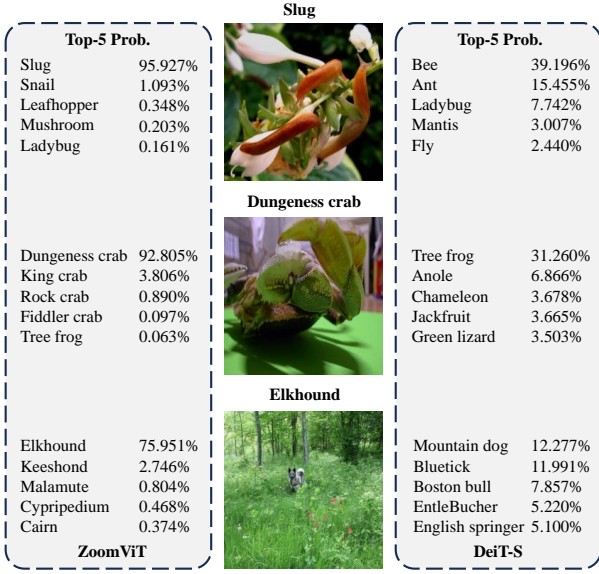

Figure 8: Visualization of top-5 classification results from both ZoomViT and DeiT.

In Figure 8 , we display the top-5 classification results from both ZoomViT and DeiT. It can be observed that ZoomViT often provides the correct classification with higher confidence when dealing with obscured targets, which aligns with our expectations.

We present in Figures 14 and 15 the input image (a), the key region score map (c) generated by the Zoomer, and the patch maps (b) under different $\alpha$ thresholds. As shown, the Zoomer first predicts potential key regions in the image, and the predicted heatmap guides the patchify process to generate patches of different sizes to achieve zoom of local key regions. From the figure, we can see that the Zoomer accurately identifies key regions in the image and generates corresponding key region score maps. In the score map (c), brighter areas (higher scores) indicate regions the model considers more important, which typically correspond to main targets or regions with significant semantic information in the image. As the $\alpha$ threshold changes, the patch map (b) exhibits different patch division strategies. When $\alpha$ is small, more regions are considered key regions, resulting in more small-sized patches to capture fine-grained local features. When $\alpha$ increases, only the highest-scoring core regions are retained as key regions, while other regions are processed with larger-sized patches, ensuring fine modeling of key regions while reducing computational overhead. This adaptive patch division mechanism demonstrates ZoomViT's core advantage: by dynamically adjusting the resolution of different regions, the model can perform more detailed analysis of important re-

gions while maintaining computational efficiency. This zoom approach mimics the human visual attention mechanism, prioritizing the most valuable information in the image, thus achieving a balance between performance and efficiency.

To further demonstrate the distinct advantages of our proposed Visual Intent-Guided paradigm, we provide a detailed conceptual and empirical comparison with CF-ViT, a representative state-of-the-art method that also employs a coarse-to-fine strategy. While both methods aim to reduce redundancy, they differ fundamentally in their execution pattern and refinement signal, leading to different behaviors in complex scenarios. The core qualitative difference lies in the guidance signal. CF-ViT relies on the Class Attention map from the coarse stage. This signal is inherently entangled with the classifier's preliminary prediction. If the coarse classifier focuses on a dominant but irrelevant object (e.g., a person instead of the held object), the refinement step will reinforce this error by zooming into the wrong region. Figure 9 visualizes the difference between CF-ViT (left) and ZoomViT (right) on challenging multi-object scenes:

- Row 1 (Tench): The image features a boy holding a fish. CF-ViT is distracted by the highly salient human face and body, failing to refine the actual target. In contrast, ZoomViT assigns a significantly higher heatmap value directly to the fish. This precise intent guidance compels the model to allocate high-resolution patches specifically to the "Tench" region, ensuring the defining features are captured for correct classification.

- Row 2 (Loafer): The scene is dominated by a man sitting on a bench, which causes CF-ViT to focus attention on the person's upper body. ZoomViT, however, correctly identifies the semantic focus. The Zoomer generates a concentrated high-value hotspot on the shoes, prioritizing high-resolution representation for the "Loafer" class while leaving the less relevant upper body in lower resolution.

- Row 3 (Bernese Mountain Dog): The target dog is positioned next to a woman, acting as a strong visual distractor. CF-ViT's attention is effectively "hijacked" by the human figure. Conversely, ZoomViT successfully distinguishes the primary subject from the distractor; it assigns maximal heatmap intensity specifically to the dog. This ensures that the animal receives dense tokenization and high-resolution processing, enabling correct recognition despite the presence of the person.

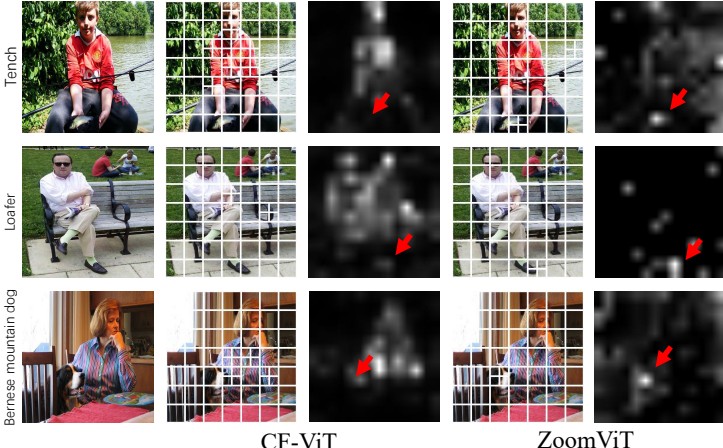

CF-ViT                ZoomViT

Figure 9: Visualization of CF-ViT (left) vs. ZoomViT (right) on challenging scenes.

## I    ERROR ANALYSIS

To better understand the limitations of ZoomViT and identify potential improvement directions, we conducted a comprehensive error analysis on misclassified samples from the ImageNet-1k validation set. As illustrated in Figure 11, we visualized typical failure cases and categorized the classification

errors into distinct scenarios based on their underlying causes. Our analysis reveals that approximately 15% of errors stem from Zoomer failures, while 85% originate from the ViT backbone component.

## I.1 ZOOMER-RELATED ERRORS

The Zoomer occasionally fails to accurately identify class-decisive regions in challenging scenarios. As shown in Figure 11(a), extremely small objects pose a significant challenge where the main subject occupies a minimal portion of the image, causing the Zoomer to inadvertently focus on background elements rather than the target object. Figure 11(b) demonstrates failures in highly complex backgrounds containing multiple visually similar objects, where the attention mechanism becomes confused by competing visual elements. Additionally, Figure 11(c) reveals cases with incorrect ground truth labels, where the Zoomer correctly identifies salient regions but the annotation itself is problematic.

## I.2 VIT BACKBONE ERRORS

More significantly, the majority of classification errors originate from the ViT backbone itself, even when the Zoomer successfully identifies the correct regions. Figure 11(d) illustrates severe obstruction scenarios where the zoomed regions provide incomplete object information due to significant occlusion, making accurate classification challenging even with proper attention guidance. Figure 11(e) shows the model's difficulty with fine-grained distinctions between visually similar categories, where the zoomed regions contain the correct objects but lack sufficient discriminative features to distinguish between closely related classes.

## J    LIMITATIONS

Although ZoomViT shows significant improvements in accuracy, particularly when handling complex images with multiple labels and occluded objects, its computational efficiency still requires further optimization. ZoomViT demonstrates excellent performance with most samples; however, we observed that some images contain multiple labels, making it difficult to categorize them into a specific single class based on guided semantics. In other words, there are errors or incomplete annotations in current large-scale image datasets, making it crucial and challenging to address these issues. Additionally, the interpretability of ZoomViT remains limited, especially in understanding how the model focuses on key areas and makes classification decisions. Enhancing interpretability is essential for building trust in model predictions, particularly in critical application domains.

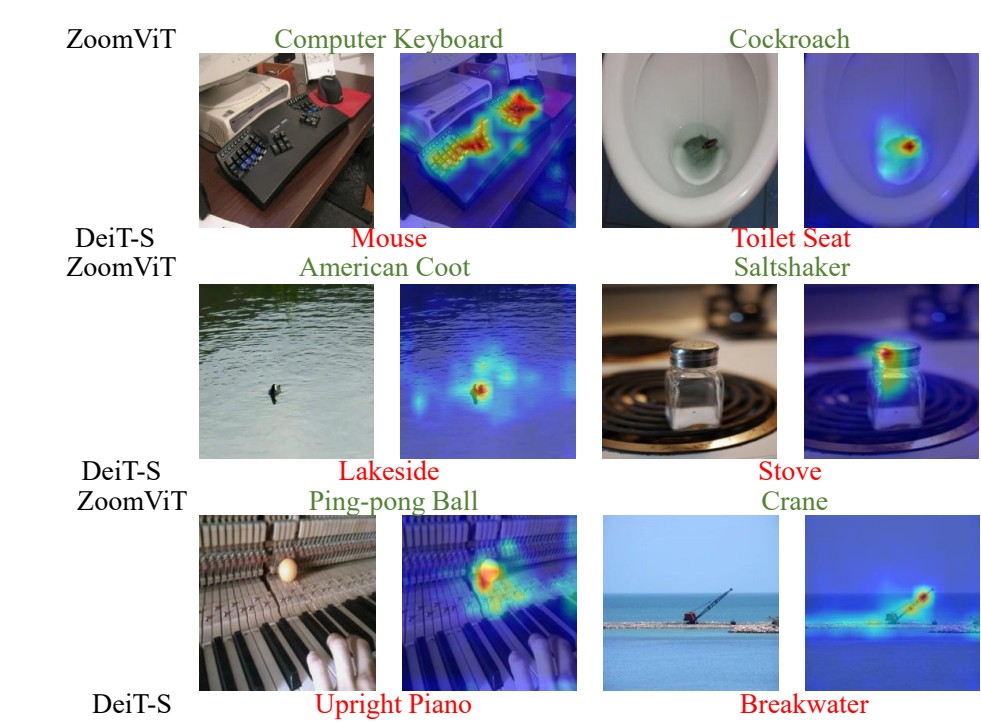

Figure 10: Visualization of class-decisive map predicted by Zoomer. A hotter area means that Zoomer predicts that objects in this area are more important.

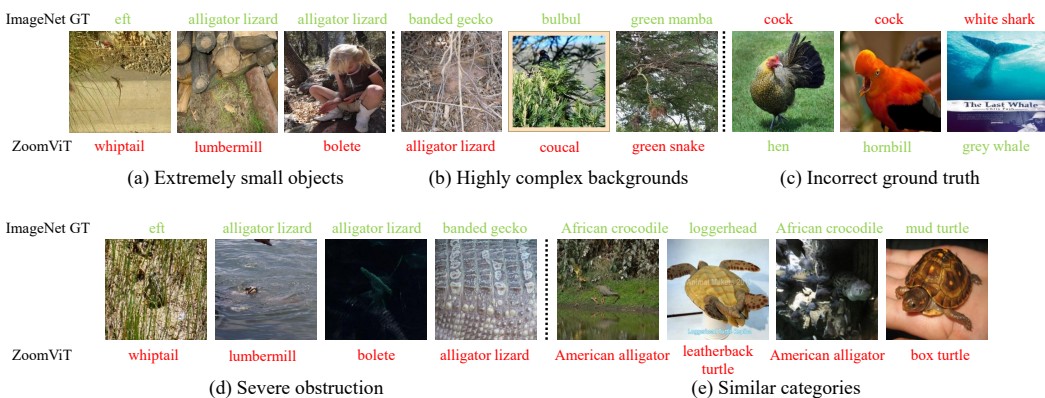

Figure 11: Error analysis of ZoomViT classification failures. (a) Extremely small objects where the Zoomer fails to focus on the tiny target. (b) Highly complex backgrounds causing attention mechanism confusion. (c) Incorrect ground truth labels where the Zoomer identifies correct regions but annotations are problematic. (d) Severe obstruction scenarios where ViT backbone struggles despite correct attention guidance. (e) Similar categories where fine-grained distinctions are challenging for the backbone network.

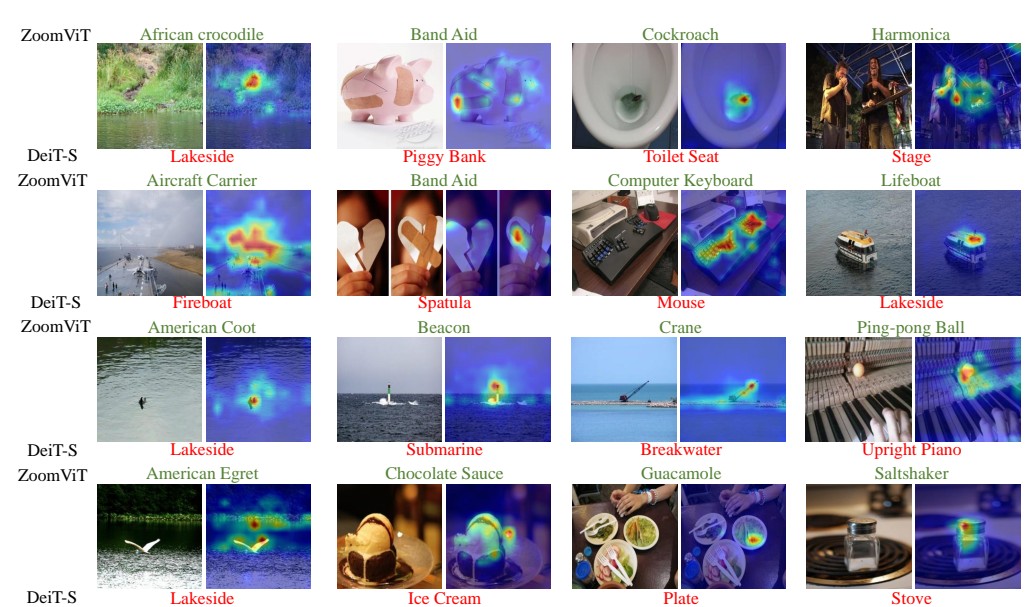

Figure 12: ZoomViT vs. DeiT-S classification. Green text (ZoomViT) indicates correct classes; red text (DeiT-S) shows incorrect ones. Zoomer helps ZoomViT identify hidden objects effectively.

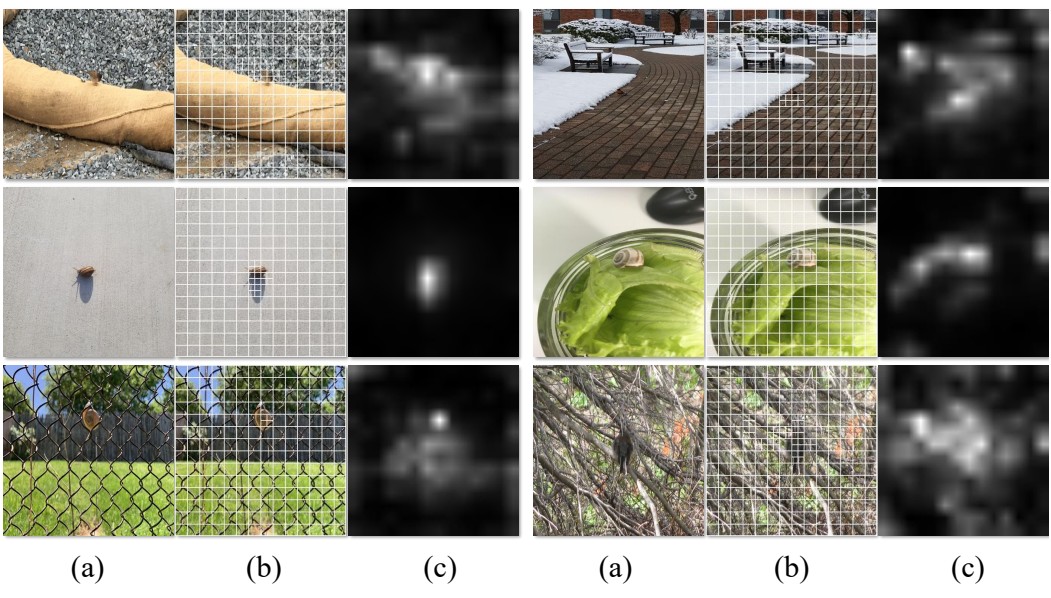

Figure 13: Intermediate process results of ImageNet-A validation set images during ZoomViT inference.

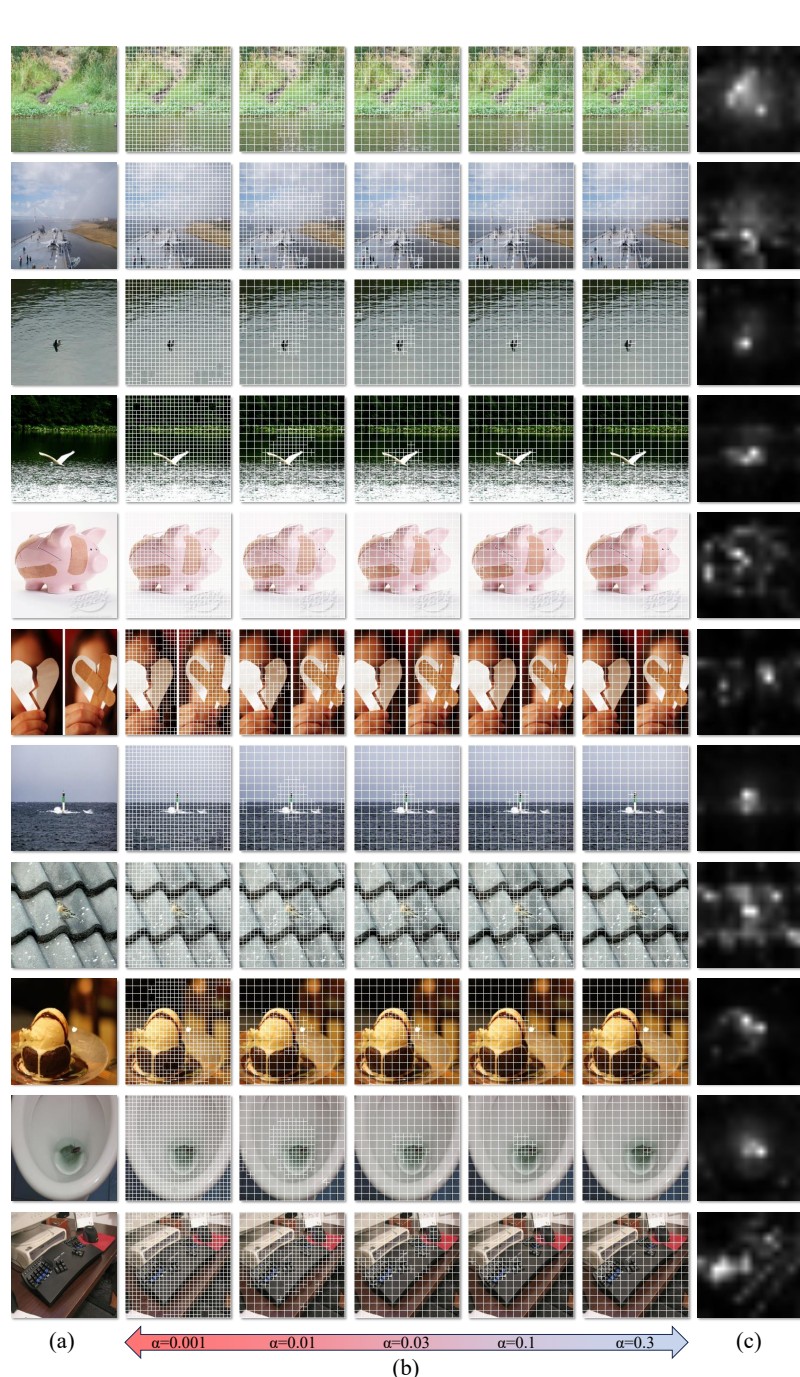

Figure 14: (**Zoom in for a clearer view**) Visualization of the input image (a), the focus region score map generated by Zoomer (c), and the segmentation map under different $\alpha$ thresholds (b).

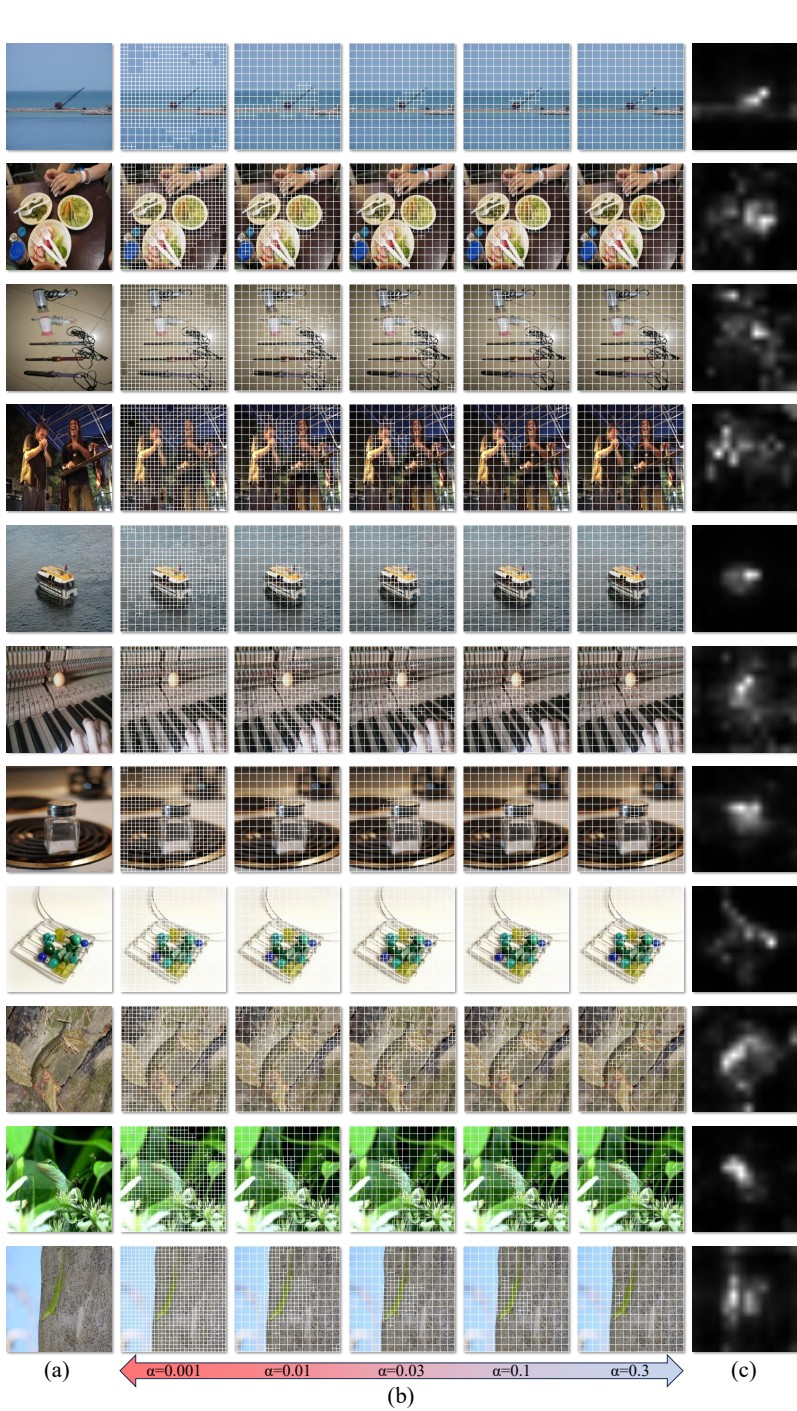

Figure 15: (**Zoom in for a clearer view**) Visualization of the input image (a), the focus region score map generated by Zoomer (c), and the segmentation map under different $\alpha$ thresholds (b).

