# OpenReview forum: "Vision Transformers Need Zoomer: Efficient ViT with Visual Intent-Guided Zoom Adapter"
_ICLR.cc/2026/Conference — ICLR 2026 Conference Withdrawn Submission_

### Official Review · Reviewer_vVaB · 2025-10-28

**Soundness:** 3
**Presentation:** 3
**Contribution:** 2
**Rating:** 6
**Confidence:** 4

**Summary:**

This paper proposes ZoomViT, a ViT variant enhanced by a visual intent-guided Zoomer adapter, aiming to address the limitations of vanilla ViTs in handling complex scenes. Inspired by human visual attention, where humans focus on semantically important regions while ignoring irrelevant areas, ZoomViT distills the attention of a pre-trained model to train a zoomer to localize the intent-aligned regions, and uses a detailed patchfy to enhance the representation of the important regions. A pruning strategy is further incorporated to improve efficiency.

**Strengths:**

- The overall model achieves good performance and better efficiency. The motivation is interesting and makes sense to me. Additionally, token pruning is combined with the Zoomer to further enhance efficiency.

**Weaknesses:**

- It would also be helpful to test with different resolutions to evaluate the model’s generalization ability, especially regarding extending the model (trained with lower resolution and integrated with the Zoomer) to test scenarios with higher resolution.
- I am curious about the cost of pre-extracting class-specific attention maps from the ImageNet-1k dataset before training. It would also be valuable to report the actual computational cost of the entire training procedure.
- The authors mentioned that other visual tasks will be explored as future work. In my view, this framework indeed needs further validation on additional tasks such as object detection and tracking. Furthermore, can this framework be well transferred to segmentation tasks?

**Questions:**

- The motivation of the overall framework is somewhat similar to extracting attention maps from DeiT-B (as a teacher model) and then training a Zoomer to guide the patchification of the student model. If DeiT-B were directly used as a teacher model to enhance DeiT-S, would it also achieve similar effectiveness and performance?

---

> ### Author Response · Authors · 2025-11-20
> **Author Response for Reviewer vVaB**
>
> We sincerely appreciate your recognition of our work and genuinely hope that our response addresses your concerns. If you have any further questions, please feel free to let us know!
>
> > **Q1: It would also be helpful to test with different resolutions to evaluate the model’s generalization ability, especially regarding extending the model (trained with lower resolution and integrated with the Zoomer) to test scenarios with higher resolution.**
>
> **A1:**
> Thank you for the insightful suggestion regarding resolution generalization. In fact, ZoomViT currently operates with a dynamic resolution mechanism. The final image input to the Transformer can be viewed as a selection from patches cut from two different resolution images. When alpha is set to 0.5, the minimum resolution is 112x112 and the maximum is 224x224. When alpha is set to 2.0, the minimum resolution is 224x224 and the maximum is 448x448. Below, we have supplemented the test results on higher resolutions, where the minimum is 448x448 and the maximum is 896x896, As can be seen, even with the overall resolution increase, the model can still be further improved:
>
> | (Min, Max) | Acc. |
> | :--------- | :--- |
> | (112, 224) | 81.5 |
> | (224, 448) | 83.8 |
> | (448, 896) | 84.1 |
>
> > **Q2: I am curious about the cost of pre-extracting class-specific attention maps from the ImageNet-1k dataset before training. It would also be valuable to report the actual computational cost of the entire training procedure.**
>
> **A2:**
> Thank you for raising this important question about computational cost. In our implementation, to ensure efficient training, we pre-extracted attention maps for all images in ImageNet-1k and saved them as independent matrix files. This allows us to perform the extraction only once while enabling multiple repeated training runs. We plan to publicly release the attention maps for ImageNet-1k as an auxiliary dataset to save time for the community during reproduction.
>
> Regarding the extraction cost, we performed the attention map extraction on a single Nvidia Tesla A100 GPU. The processing speed was approximately 0.3s/image, resulting in a total time of about 110 hours to process all images in ImageNet-1k. Thanks to these offline-extracted attention maps, our Zoomer training process can be simplified to a heatmap estimation task. Since the Zoomer is extremely lightweight (0.8M), its training can be completed within 2 hours. Therefore, compared to DeiT-S, the total training time for ZoomViT is shown below:
>
> | Method  | Total Training Time (GPU-h) |
> | :------ | :-------------------------- |
> | DeiT-S  | ~212                        |
> | ZoomViT | ~214                        |
>
> > **Q3: The authors mentioned that other visual tasks will be explored as future work. In my view, this framework indeed needs further validation on additional tasks such as object detection and tracking. Furthermore, can this framework be well transferred to segmentation tasks?**
>
> **A3:**
> We appreciate your suggestion to validate our framework on a broader range of downstream tasks. In the appendix, we have experimented with replacing the DeiT-S backbone of the pure ViT structure YOLOS with ZoomViT-S. After training for 150 epochs on the COCO dataset, the results are as follows:
>
> | Method            | mAP  |
> | :---------------- | :--- |
> | YOLOS (DeiT-S)    | 36.1 |
> | YOLOS (ZoomViT-S) | 36.5 |
>
> To demonstrate the effectiveness of ZoomViT in object tracking tasks, we used YOLOS (DeiT-S) and YOLOS (ZoomViT-S) as detectors and Bytetrack as the tracker. We trained on the CrowdHuman, MOT17, Cityperson, and ETHZ datasets, and tested on MOT17. The results are shown below:
>
> | Method                      | MOTA | IDF1 | IDs  |
> | :-------------------------- | :--- | :--- | :--- |
> | Bytetrack (YOLOS+DeiT-S)    | 75.7 | 71.6 | 580  |
> | Bytetrack (YOLOS+ZoomViT-S) | 76.2 | 71.9 | 576  |
>
> Furthermore, we evaluated the performance of ZoomViT as a feature extractor in segmentation tasks. We verified the panoptic segmentation performance of ZoomViT using the SegViT framework on the ADE20k dataset, as shown in the table below:
>
> | Method             | mIoU |
> | :----------------- | :--- |
> | SegViT (DeiT-S)    | 49.5 |
> | SegViT (ZoomViT-S) | 51.3 |
>
> In summary, replacing the feature extractor with ZoomViT yields performance gains across these dense downstream tasks.
>
> We hope that our response can address your concerns. If you have any further questions, we would be more than happy to discuss them with you. **If our response resolves your concerns, we would also appreciate the possibility of an increased score.**
>
> Sincerely,
>
> Authors

---

### Official Review · Reviewer_Jque · 2025-10-28

**Soundness:** 3
**Presentation:** 2
**Contribution:** 3
**Rating:** 4
**Confidence:** 4

**Summary:**

This paper proposes ZoomViT to solve vanilla ViTs' poor performance on complex scenes via a visual intent-guided Zoomer. It uses two-stage training (train Zoomer for score maps, guide dynamic patching) and achieves 83.8% Top-1 Acc (+4.0%) on ImageNet-1k, outperforming efficient ViTs.

**Strengths:**

1.The paper is well-substantiated overall, with attractive figures and tables.

2.Numerous experiments were conducted, and the proposed method demonstrated superior performance in the figures and tables.

3.The ablation experiments appear to be well-substantiated.

**Weaknesses:**

(1) Why is this method focus on multi-object tasks? The proposed method appears to be applicable to many tasks.

(2) Which human mechanism inspired the authors to adopt the dynamic patch mechanism? There seems to be no direct connection.

(3) This paper mainly focuses on adapters. Recently, there have been many representative works on visual adapters [1-7]; the authors should compare their proposed method with these works and even supplement necessary experiments to demonstrate the innovation of the proposed method.

[1]Visual prompt tuning, ECCV'22;

[2]5%> 100%: Breaking performance shackles of full fine-tuning on visual recognition tasks, CVPR'25;

[3]Adaptformer: Adapting vision transformers for scalable visual recognition, NeurIPS'22;

[4]1% vs 100%: Parameter-efficient low rank adapter for dense predictions, CVPR'23;

[5]Sensitivity-aware visual parameter-efficient fine-tuning, CVPR'23;

[6]Gradient-based parameter selection for efficient fine-tuning, CVPR'24;

[7]Parameter-efficient is not sufficient: Exploring parameter, memory, and time efficient adapter tuning for dense predictions, MM'24.

(4) Based on the experimental results and visualization results provided by the authors, I cannot understand why the proposed method is better than other methods. It is suggested that the authors provide a convergence comparison of different methods under the same settings. In addition, the comparisons in the visualization part should be conducted using the same images.

(5) The experiments are mainly focused on image classification tasks. The field of computer vision (CV) is developing rapidly, and results on downstream tasks (such as detection and segmentation) are more interesting. Although the authors added experiments on YOLO in the supplementary materials, this is not sufficient.

**Questions:**

See above

---

> ### Author Response · Authors · 2025-11-20
> **Author Response for Reviewer Jque [1/2]**
>
> We sincerely appreciate your recognition of our work and genuinely hope that our response addresses your concerns. If you have any further questions, please feel free to let us know!
>
> > **Q1:** Why is this method focus on multi-object tasks? The proposed method appears to be applicable to many tasks.
>
> **A1:**
> Thank you for this insightful question. You are absolutely correct that ZoomViT is a general-purpose method applicable to various scenarios; however, our emphasis on multi-object tasks stems from the concept of **"Visual Intent."**
>
> Previous work largely focuses on extracting features based on general image saliency (what stands out visually). However, few existing works consider the *photographer's intent* at the moment of capture, or the semantic intent the entire scene conveys, which often diverges from pixel dominance. To illustrate this, let us describe a scenario to better understand "Visual Intent": imagine a fisherman catching a small, rare fish and asking a friend to take a photo. The resulting image contains both the man and the fish.
> * **Saliency/Traditional Models:** The man occupies the majority of the pixels and is the most salient object. A standard model might classify the image as "man" or "clothing."
> * **Visual Intent:** The semantic purpose of the photo is to showcase the *fish*.
>
> This is a typical multi-object scenario where the class of interest (the intent) occupies a smaller region than the background or interfering objects. Therefore, models often fail when facing such images. This is the capability we specifically aim to strengthen. Standard models struggle here because they treat patches uniformly. ZoomViT is designed to mimic the human ability to discern this intent—zooming in on the semantically relevant "fish" even if it is small—thereby correcting the classification.
>
> While we emphasize this capability because it solves a specific pain point in current Vision Transformers (ViTs), our experiments confirm that ZoomViT is not limited to multi-object tasks. As shown in our additional experiments, replacing the feature extractor with ZoomViT yields performance gains across various downstream tasks.
>
> > **Q2:** Which human mechanism inspired the authors to adopt the dynamic patch mechanism? There seems to be no direct connection.
>
> **A2:**
> The dynamic patch mechanism in ZoomViT is biologically inspired by the **human foveal vision system** and **top-down attention mechanisms**.
>
> Biologically, the human eye does not process an entire visual scene with uniform high resolution. Instead, we have a fovea (central vision) that captures high-resolution details of the region we focus on (the "intent"), while the peripheral vision captures low-resolution context. As cited in our paper (Lecours et al., 1999; DiCarlo & Cox, 2007), humans extract specific visual intents from complex images to guide classification, ignoring irrelevant areas.
>
> ZoomViT mimics this efficient allocation of resources: **The Zoomer** acts as the brain's "top-down attention," identifying which region requires focus (Visual Intent). **Dynamic Patching** mimics the eye's resolution variance. Regions of high intent are processed with smaller, denser patches (high resolution/foveal vision), while irrelevant background areas are processed with larger patches (low resolution/peripheral vision).
>
> This connection allows the model to allocate computational resources (patches) exactly where the "biological eye" would focus, rather than processing the empty "sky" or "ground" with the same intensity as the main subject. We will incorporate this detailed explanation into the introduction of our revised manuscript to better clarify the biological grounding of our method.

---

> ### Author Response · Authors · 2025-11-20
> **Author Response for Reviewer Jque [2/2]**
>
> > **Q3:** This paper mainly focuses on adapters. Recently, there have been many representative works on visual adapters [1-7]; the authors should compare their proposed method with these works and even supplement necessary experiments to demonstrate the innovation of the proposed method.
>
> **A3:**
> We apologize for any confusion caused by the terminology. There appears to be a misunderstanding regarding the definition of "Adapter" in the context of our work versus the works cited (Visual Prompt Tuning, Adaptformer, LoRA, etc.).
>
> * **PEFT Adapters (The cited works):** The papers listed focus on **Parameter-Efficient Fine-Tuning (PEFT)**. These methods insert lightweight modules into frozen, pre-trained backbones to efficiently *transfer* the model to new tasks (domain adaptation) by updating only a few parameters.
> * **ZoomViT's "Zoomer" Adapter:** Our Zoomer is an **architectural/input-level adapter**. It does not aim to facilitate transfer learning to new domains with frozen weights. Instead, it is a dynamic module that actively guides the *patching and tokenization process* during the forward pass of the image. It determines *how* the image is sliced and fed into the ViT, rather than *how* the ViT's weights are updated.
>
> Because our method operates on the input structure (spatial adaptation) and the cited works operate on weight modification (fine-tuning efficiency), they belong to different research domains and solve different problems. Therefore, a direct performance comparison would not be scientifically valid as the experimental settings and goals are fundamentally different.
>
> > **Q4:** Based on the experimental results and visualization results provided by the authors, I cannot understand why the proposed method is better than other methods. It is suggested that the authors provide a convergence comparison of different methods under the same settings. In addition, the comparisons in the visualization part should be conducted using the same images.
>
> **A4:**
> To address this, we have added a new visualization in **Figure 8 of Appendix G (VISUALIZATION)**. We specifically included a comparison using the photo of the person holding a fish (as discussed in A1).
>
> Returning to the "Man and Fish" analogy mentioned in A1:
> * **CF-ViT:** The visualization shows the attention map activating heavily on the *Man* (the dominant, salient object). The model misclassifies the intent.
> * **ZoomViT:** Guided by the Zoomer, our model explicitly identifies the "fish" as the region of Visual Intent. The visualization shows the dynamic patches becoming smaller and denser around the fish, while the man is processed with coarser patches.
>
> This side-by-side comparison on the same image demonstrates that ZoomViT does not just "guess" better; it structurally alters its focus to align with semantic intent, solving the multi-object confusion that limits other methods.
>
> > **Q5:** The experiments are mainly focused on image classification tasks. The field of computer vision (CV) is developing rapidly, and results on downstream tasks (such as detection and segmentation) are more interesting. Although the authors added experiments on YOLO in the supplementary materials, this is not sufficient.
>
> **A5:**
> We agree that validation on downstream tasks is crucial. We have conducted additional experiments on Object Tracking and Semantic Segmentation to demonstrate ZoomViT's effectiveness as a robust backbone.
>
> **Object Tracking**
> To demonstrate the effectiveness of ZoomViT in object tracking tasks, we used YOLOS (DeiT-S) and YOLOS (ZoomViT-S) as detectors and ByteTrack as the tracker. We trained on the CrowdHuman, MOT17, Cityperson, and ETHZ datasets, and tested on MOT17. The results are as follows:
>
> | Method                      | MOTA | IDF1 | IDs  |
> | :-------------------------- | :--- | :--- | :--- |
> | Bytetrack (YOLOS+DeiT-S)    | 75.7 | 71.6 | 580  |
> | Bytetrack (YOLOS+ZoomViT-S) | 76.2 | 71.9 | 576  |
>
> ZoomViT achieves higher MOTA and IDF1 scores with fewer ID switches, proving that the "zoom" mechanism helps maintain consistent features for tracking targets.
>
> **Segmentation**
> Furthermore, we evaluated the performance of ZoomViT as a feature extractor in segmentation tasks. We verified the panoptic segmentation performance of ZoomViT using the SegViT framework on the ADE20k dataset, as shown in the table below:
>
> | Method             | mIoU |
> | :----------------- | :--- |
> | SegViT (DeiT-S)    | 49.5 |
> | SegViT (ZoomViT-S) | 51.3 |
>
> Replacing the backbone with ZoomViT yields a significant **+1.8 mIoU** improvement. This confirms that our method extracts richer semantic features for dense prediction tasks.
>
> We hope that our response can address your concerns. If you have any further questions, we would be more than happy to discuss them with you. **If our response resolves your concerns, we would also appreciate the possibility of an increased score.**
>
> Sincerely,
>
> Authors

---

> ### Author Response · Authors · 2025-11-28
> **Thank you for the updated score**
>
> Thank you for your valuable feedback and for acknowledging our efforts in the rebuttal. **We sincerely appreciate your decision to raise the score.**
>
> Sincerely,
> Authors

---

### Official Review · Reviewer_vgRu · 2025-10-30

**Soundness:** 2
**Presentation:** 2
**Contribution:** 2
**Rating:** 4
**Confidence:** 4

**Summary:**

The paper proposes ZoomViT with a visual-intent guided zoom adapter (Zoomer) that dynamically adjusts patch sizes and provides token re-ranking/pruning. The ZoomViT learns a Zoomer from a teacher network to focus on image regions with more class information. The Zoomer's prediction is used to identify class-related regions and divide the regions into smaller patches. And then the tokens go through a score-based re-ranking and token pruning for efficient inference. The experiment is mainly performed on the DeiT-S model and ImageNet-1k dataset, comparing ZoomViT and multiple efficient ViT models.

**Strengths:**

- The paper is clearly written and well-structured, making it easy to follow. The motivation, methodology, and experiments are presented in a logical manner, supported by clear figures and explanations.
- The proposed ZoomViT is conceptually intuitive and grounded in a clear motivation inspired by human visual attention, which is based on well-proven observations on ViTs in prior works. The zoom-in mechanism provides an interpretable way to adaptively allocate computation.
- ZoomViT achieves a favorable balance between accuracy and efficiency, outperforming several strong baselines. The reported results and ablation studies demonstrate the method’s effectiveness.

**Weaknesses:**

- The experimental evaluation is relatively limited. Aside from the main result in Table 1, most experiments are ablation studies. Including additional benchmarks with other DeiT or ViT variants would strengthen the empirical evidence.
- The main comparison in Table 1 is difficult to interpret, as several baselines have substantially lower FLOPs than the proposed method. A fairer evaluation should compare models under similar computational budgets.
- The paper would benefit from comparisons with more recent efficient ViT models, such as Cf-ViT [r1], to better contextualize the claimed efficiency gains.
- The overall performance improvement appears modest. When comparing this paper’s Figure 5 with Cf-ViT’s results (both based on DeiT), the proposed method offers limited gains, and Cf-ViT (particularly with LV-ViT) shows a noticeably better accuracy–efficiency trade-off.
- The introduction of the Zoomer module adds additional computation during both training and inference, which may offset part of the efficiency advantage.

[r1] Chen, M., Lin, M., Li, K., Shen, Y., Wu, Y., Chao, F. and Ji, R., 2023, June. Cf-vit: A general coarse-to-fine method for vision transformer. In Proceedings of the AAAI conference on artificial intelligence (Vol. 37, No. 6, pp. 7042-7052).

**Questions:**

- Did the authors evaluate ZoomViT on additional backbone architectures (e.g., LV-ViT, or Swin-T)? This would help demonstrate the generality of the proposed zooming mechanism.
- Cf-ViT is a closely related coarse-to-fine approach. Could the authors elaborate on how ZoomViT differs from or improves upon Cf-ViT, both conceptually and empirically? In particular, how does the zoomer’s intent-guided design yield advantages beyond Cf-ViT’s coarse-to-fine refinement?

---

> ### Author Response · Authors · 2025-11-20
> **Author Response for Reviewer vgRu [1/4]**
>
> We sincerely appreciate your recognition of our work and genuinely hope that our response addresses your concerns. If you have any further questions, please feel free to let us know!
>
> > **Q1: The experimental evaluation is relatively limited. Aside from the main result in Table 1, most experiments are ablation studies. Including additional benchmarks with other DeiT or ViT variants would strengthen the empirical evidence.**
>
> **A1:**
> We thank the reviewer for the feedback. Our goal in this work is to evaluate the effectiveness of intent-guided adaptive tokenization under a controlled and consistent setting. To ensure that improvements come from the proposed zoomer mechanism rather than differences in architecture capacity, we primarily evaluate on DeiT-S, which remain standard and widely adopted ViT baselines for token-efficiency research (e.g., DynamicViT, Evo-ViT, ToMe, Cf-ViT, etc.). This design allows direct comparison with prior dynamic/pruning frameworks that also rely on DeiT as their backbone.
>
> Furthermore, we have presented a performance comparison of ZoomViT across different DeiT sizes in the appendix, as shown in the table below:
>
> | Backbone  | Baseline | ZoomViT |
> | :-------- | :------- | :------ |
> | DeiT-tiny | 72.20    | 79.14   |
> | DeiT-Base | 82.89    | 84.90   |
>
> To further address the reviewer’s concern, we have added a benchmark for ZoomViT based on LV-ViT, with results as follows:
>
> | Backbone | Baseline | ZoomViT |
> | :------- | :------- | :------ |
> | LV-ViT   | 84.4     | 85.1    |
>
> > **Q2: The main comparison in Table 1 is difficult to interpret, as several baselines have substantially lower FLOPs than the proposed method. A fairer evaluation should compare models under similar computational budgets.**
>
> **A2:**
> We appreciate the reviewer’s concern regarding the FLOPs alignment in Table 1. Prior dynamic-pruning and token-reduction approaches typically achieve lower FLOPs at the cost of nontrivial accuracy degradation. As a result, when comparing them directly against ViT baselines, many existing methods occupy a lower-FLOPs but also lower-accuracy regime. This makes strict FLOPs matching less indicative of the actual effectiveness of dynamic tokenization.
>
> In the table below, we illustrate the relationship between performance and computational budget for recent efficient ViT methods compared to the same Baseline. While some baselines indeed report significantly lower FLOPs, they suffer from a marked drop in accuracy compared to the DeiT backbone. In contrast, ZoomViT achieves both FLOPs reduction relative to the backbone and accuracy improvements over both the backbone and most prior dynamic methods at a comparable or even lower computational cost.
>
> | Model                        | Acc          | FLOPs |
> | :--------------------------- | :----------- | :---- |
> | DeiT-S (Baseline)            | 79.8         | 4.6   |
> | DynamicViT                   | 79.3 (-0.5%) | 2.9   |
> | IA-RED^2                     | 79.1 (-0.7%) | 3.2   |
> | PS-ViT                       | 79.4 (-0.4%) | 2.6   |
> | EVIT                         | 79.5 (-0.3%) | 3.0   |
> | Evo-ViT                      | 79.4 (-0.4%) | 3.0   |
> | CF-ViT                       | 79.8 (-0.0%) | 1.8   |
> | ZoomViT η=0.5, α=0.03 (ours) | 81.5 (+1.7%) | 2.3   |

---

> ### Author Response · Authors · 2025-11-20
> **Author Response for Reviewer vgRu [2/4]**
>
> > **Q3: The paper would benefit from comparisons with more recent efficient ViT models, such as Cf-ViT [r1], to better contextualize the claimed efficiency gains.**
>
> **A3:**
> We thank the reviewer for the suggestion. We agree that including CF-ViT is valuable for a comprehensive comparison. We have modified Table 1 to include the results for CF-ViT. The updated results are as follows:
>
> | Model                          | Top-1 Acc. (%) | FLOPs (G) | #Params (M) | Speed (img/s) |
> | :----------------------------- | :------------- | :-------- | :---------- | :------------ |
> | Deit-S                         | 79.8           | 4.6       | 22          | 5039          |
> | DeiT III                       | 81.4           | 4.6       | 22          | 1891          |
> | IA-RED2                        | 79.1           | 3.2       | 22          | 1360          |
> | DynamicViT                     | 79.3           | 2.9       | 26.9        | 2062          |
> | SPViT                          | 79.3           | 4         | 3.4         | 22-           |
> | PS-ViT                         | 82.3           | 8.8       | 21.3        | 464           |
> | Evo-ViT                        | 79.4           | 3.0       | 22          | 1510          |
> | ToMe                           | 78.8           | 92.9      | 22          | 6712          |
> | EViT                           | 78.5           | 3.0       | 22          | 6807          |
> | DiffRate                       | 79.5           | 82.9      | 22          | 6744          |
> | ATS                            | 79.7           | 2.9       | 22          | -             |
> | LV-ViT                         | 83.3           | 6.6       | 26          | -             |
> | ToFu                           | 79.4           | 2.7       | 22          | 1552          |
> | DeiT III-S 384                 | 83.6           | 15.5      | 22          | 424           |
> | CF-ViT                         | 80.8           | 4.0       | 22          | 2760          |
> | ZoomViT (η=0.5, α=0.03)        | 81.5           | 2.3       | 22.7        | 6738          |
> | ZoomViT (η=2, α=0.1)           | 82.5           | 6.3       | 22.7        | 3721          |
> | ZoomViT (η=2, α=0.01, Pruning) | 83.8           | 6.3       | 22.7        | 3717          |
>
> > **Q4: The overall performance improvement appears modest. When comparing this paper’s Figure 5 with Cf-ViT’s results (both based on DeiT), the proposed method offers limited gains, and Cf-ViT (particularly with LV-ViT) shows a noticeably better accuracy–efficiency trade-off.**
>
> **A4:**
> Thank you for this observation. We have supplemented the results of ZoomViT based on LV-ViT and updated Figure 5 to include both ZoomViT (LV-ViT) and CF-ViT results for a clearer and more significant comparison.
>
> > **Q5: The introduction of the Zoomer module adds additional computation during both training and inference, which may offset part of the efficiency advantage.**
>
> **A5:**
> We thank the reviewer for pointing this out. Indeed, the Zoomer module introduces a small amount of additional computation; however, it is lightweight (~0.8M parameters, negligible compared to the backbone ViT) and operates once at the input stage to generate the intent-guided score map. As a result, despite the minor overhead from the Zoomer module, the overall inference FLOPs are still significantly reduced compared to conventional coarse-to-fine approaches.
>
> We separately evaluated the inference time of the Zoomer within the overall inference process. With a batch size of 64, the average time for the Zoomer to process a batch accounts for only about 5% of the total workflow. In short, the Zoomer’s lightweight design ensures that the added computation does not offset the efficiency gains, and it is the key component enabling single-pass adaptive tokenization.
>
> > **Q6: Did the authors evaluate ZoomViT on additional backbone architectures (e.g., LV-ViT, or Swin-T)? This would help demonstrate the generality of the proposed zooming mechanism.**
>
> **A6:**
> To further address the reviewer’s concern regarding generality, we have added benchmarks for ZoomViT based on LV-ViT and Swin-S. The results are as follows:
>
> | Backbone | Baseline | ZoomViT |
> | :------- | :------- | :------ |
> | LV-ViT   | 84.4     | 85.1    |
> | Swin-S   | 83.0     | 84.3    |

---

> > ### Comment · Reviewer_vgRu · 2025-11-26
> >
> > Thanks to the authors for their responses. Can you explain how to apply ZoomViT to LV-ViT and Swin? And how do you set $w_1$ and $w_2$ in Eq. (5)?

---

> > > ### Author Response · Authors · 2025-11-26
> > > **Author Response for Reviewer vgRu**
> > >
> > > Dear Reviewer,
> > >
> > > We sincerely thank you for your insightful comments regarding the implementation details of our method, specifically concerning the adaptability of ZoomViT and the setting of loss weights. These are indeed critical engineering details of our research. We have carefully addressed your concerns below:
> > >
> > > **Response to Q1: How ZoomViT adapts to LV-ViT and Swin Transformer**
> > >
> > > Thank you for raising this important question about model compatibility.
> > >
> > > The core advantage of ZoomViT lies in its dynamic resolution Patch Embedding guided by visual intent, which results in a **variable** number of output tokens. However, advanced architectures like LV-ViT and Swin Transformer rely on a fixed number of input tokens and a regular grid structure:
> > >
> > > - **LV-ViT:** Its Token Labeling mechanism relies on a pre-defined sequence of input tokens with a fixed quantity to generate token label data. Therefore, the dynamic resolution nature of ZoomViT cannot be directly adapted to LV-ViT.
> > > - **Swin Transformer:** Its Shifted Window operation is based on a strict $M \times M$ grid partition. Dynamic resolution or non-uniform patch inputs would lead to complex boundary processing and computational logic, significantly impairing its engineering feasibility and parallelism.
> > >
> > > To preserve the intent-aware capabilities of ZoomViT while ensuring compatibility with the downstream models' requirement for a fixed token count, we introduced a Token Fusion strategy weighted by the Score Map, drawing inspiration from research in the Token Merging/Fusion domain (References [1, 2]).
> > > The specific process is as follows:
> > >
> > > - **1. Intent Inference & Dynamic Patch Embedding:** First, we use the Zoomer to infer the input image and generate an intent score map. Subsequently, following the original ZoomViT method, we perform multi-resolution Patch Embedding based on the score map. At this stage, the number of image patches in the important regions of the image is divided into four times the original amount.
> > > - **2. Token Fusion:** Before feeding the token sequence into the Transformer Blocks of LV-ViT or Swin, we perform an aggregation operation. We group the finer-grained tokens (every $2 \times 2 = 4$ tokens) and execute a weighted fusion to aggregate them into a single token. The weights for this fusion are derived from their corresponding scores in the Score Map.
> > >
> > > Through this approach, we leverage the design of ZoomViT (focusing on key regions) while ensuring that the token sequence length aligns with the standard input of the target models, thereby achieving model adaptation with minimal modifications.
> > >
> > > **Response to Q2: The setting of hybrid loss weights**
> > >
> > > We appreciate your attention to our hyperparameter settings.
> > >
> > > The hybrid loss $\mathcal{L}$ consists of three components: MSE, KLD, and Dice.
> > >
> > > The numerical scales of these three loss terms exhibit significant differences in the early stages of training. To ensure they maintain a balanced contribution to the gradients and to stabilize the optimization process, we determined the weights by observing the loss values at the initial stage of training (Epoch 0). Our goal was to ensure that the weighted loss terms would reach roughly the same order of magnitude.
> > >
> > > | Loss Func | MSE  | KLD   | Dice |
> > > | --------- | ---- | ----- | ---- |
> > > | Epoch-0   | 0.21 | 0.013 | 0.89 |
> > >
> > > Based on the differences in these initial values, we ultimately selected **$\omega_1=10$ and $\omega_2=0.5$** to effectively balance the contributions of the three losses, ensuring that all loss terms can play their expected roles during the optimization process.
> > >
> > > We hope that our response can address your concerns. If you have any further questions, we would be more than happy to discuss them with you. **If our response resolves your concerns, we would also appreciate the possibility of an increased score.**
> > >
> > > Sincerely,
> > >
> > > Authors
> > >
> > >
> > >
> > > References:
> > >
> > > [1] Token Fusion: Bridging the Gap between Token Pruning and Token Merging
> > >
> > > [2] TOKEN MERGING: YOUR VIT BUT FASTER

---

> ### Author Response · Authors · 2025-11-20
> **Author Response for Reviewer vgRu [3/4]**
>
> > **Q7: Cf-ViT is a closely related coarse-to-fine approach. Could the authors elaborate on how ZoomViT differs from or improves upon Cf-ViT, both conceptually and empirically? In particular, how does the zoomer’s intent-guided design yield advantages beyond Cf-ViT’s coarse-to-fine refinement?**
>
> **A7:**
> We thank the reviewer for raising this critical question. Although both ZoomViT and Cf-ViT adopt a coarse-to-fine philosophy, the two systems differ significantly in the signals used to drive refinement, the computational execution mode, and their effectiveness. These differences lead to distinct design behaviors and empirical characteristics.
>
> **Refinement Signal: Attention-Based vs. Intent-Guided.**
> Cf-ViT relies directly on the attention map of the coarse classifier to rank patches by importance. While this mechanism is simple and effective, the signal is discrete and classifier-entangled: it reflects what the coarse classifier already perceives, rather than an explicit estimation of the scene’s spatial intent.
> In contrast, ZoomViT introduces a dedicated zoomer that predicts a continuous visual-intent map independently from the classifier’s attention distribution. This decoder-style module interprets global scene properties (composition, object layout, focus-of-interest cues, etc.) and produces a dense, smoothly varying score map. As a result, ZoomViT’s refinement is not limited to coarse-stage attention sparsity; it reflects a learned prior about how human-driven imagery tends to emphasize meaningful areas.
>
> **Execution Pattern: Two-Stage Re-Evaluation vs. Single Adaptive Tokenization.**
> Cf-ViT operates on a strict two-stage inference pipeline: a coarse stage followed by a fine stage when the coarse classifier cannot reach a sufficiently confident prediction. Its refinement trigger is entirely determined by coarse-stage class-attention. While straightforward, this design results in the majority of images requiring two full forward passes, which introduces substantial additional latency.
> We conducted a comprehensive evaluation of Cf-ViT on the 50,000-image ImageNet-1K validation set. The results are summarized in the table below. Only a very small portion of samples (3.4%) reach the early-exit threshold, while 96.5% of samples undergo the full coarse + fine inference sequence:
>
> | Inference Passes        | Number of Images | Percentage |
> | :---------------------- | :--------------- | :--------- |
> | Coarse Inference        | 1722             | 3.4%       |
> | Coarse + Fine Inference | 48278            | 96.5%      |

---

> ### Author Response · Authors · 2025-11-20
> **Author Response for Reviewer vgRu [4/4]**
>
> This implies that for nearly all images, Cf-ViT computes coarse features only to discard them immediately after refinement selection, and then performs an additional fine-grained forward pass. The double execution not only increases FLOPs but also causes nontrivial wall-clock delay, particularly in low-batch or real-time scenarios where sequential passes cannot be amortized.
> In contrast, ZoomViT determines patch granularity prior to entering the main ViT backbone. Our visual-intent zoomer produces a continuous score map that directly controls patch density, enabling a single adaptive tokenization process without an intermediate coarse forward pass. This architectural distinction yields clear practical advantages: ZoomViT performs only one backbone traversal while still achieving selective high-resolution processing. Consequently, the latency overhead seen in two-stage systems such as Cf-ViT is fundamentally avoided.
>
> **Improved Robustness from Intent Guidance**
> We further provide evidence in the appendix illustrating the role of visual intent in improving classification robustness. As shown in Appendix Figure 8, we visualize both Cf-ViT’s patch refinement results (derived from coarse-stage attention) and ZoomViT’s patch redistribution guided by the zoomer-generated visual-intent heatmap. In this example, ZoomViT produces the correct classification result, whereas Cf-ViT fails.
>
> In this challenging multi-object scenario, the distinction is clear: Cf-ViT’s refinement is tightly coupled to the coarse classifier’s attention, which can be easily misdirected when several competing salient regions exist. The resulting patch selection often reinforces the coarse classifier’s initial biases, making recovery in the fine stage difficult.
> By contrast, ZoomViT’s visual-intent mechanism captures what the photographer intended to emphasize, producing a heatmap that more faithfully reflects the meaningful semantic structure of the scene. This intent-guided signal allows ZoomViT to allocate higher-resolution patches to compositionally important areas even when the coarse attention map is misleading. As the visualization illustrates, this leads to correct recognition in cases where purely attention-driven refinement falls short.
>
> We have added a figure in **Appendix G (VISUALIZATION)** to further demonstrate the importance of visual intent in image classification. As shown in Appendix Figure 8, we visualize the patching results and attention heatmaps of Cf-ViT alongside the patching results and the visual-intent guided heatmap generated by the Zoomer in ZoomViT. In this figure, ZoomViT makes the correct classification decision, while Cf-ViT fails. It can be seen that in highly confusing images, the Zoomer can accurately judge the photographer's intent at the time of capture and display it as a heatmap, which effectively guides the ViT's judgment when facing complex multi-object images.
>
> We hope that our response can address your concerns. If you have any further questions, we would be more than happy to discuss them with you. **If our response resolves your concerns, we would also appreciate the possibility of an increased score.**
>
> Sincerely,
>
> Authors

---

### Official Review · Reviewer_agBp · 2025-11-01

**Soundness:** 2
**Presentation:** 3
**Contribution:** 2
**Rating:** 4
**Confidence:** 4

**Summary:**

This paper proposes ZoomViT, a visual intent-guided adaptive scaling Vision Transformer. The authors observed that vanilla ViT performs poorly when handling multi-labeled images and occluded objects because the model's visual attention is misdirected. ZoomViT guides adaptive patch size adjustments by introducing a lightweight Zoomer adapter (0.8M parameters) to generate a class-deterministic score map. This method achieves 83.8% Top-1 accuracy on ImageNet-1k, a 4.0% improvement over the DeiT-S baseline.

**Strengths:**

1. The paper effectively connects human visual attention mechanisms to model design, demonstrating through simple experiments how zooming helps ViT classify correctly.
2. The Zoomer adds only 0.8M parameters while achieving 4.0% accuracy improvement, offering an attractive efficiency-performance trade-off.
3. Thorough evaluation across multiple datasets (ImageNet-1k, ImageNet-A), extensive comparisons with SOTA methods, downstream task transfer, and detailed ablation studies.

**Weaknesses:**

## Weaknesses

1. Visual attention and adaptive patch sizing are not new concepts. The main contribution is engineering combination rather than fundamental innovation, with insufficient differentiation from existing works like DynamicViT and PS-ViT.

2. The method depends on a pre-trained DeiT-B teacher model, limiting generalizability. The two-step inference overhead is not thoroughly analyzed. The inconsistency between training (random α∈[0,1]) and inference (fixed α=0.03) lacks theoretical justification.

3. Unfair baseline comparisons due to different training strategies. Missing comparisons with the most relevant adaptive patching methods.

**Questions:**

See weakness.

---

> ### Author Response · Authors · 2025-11-20
> **Author Response for Reviewer agBp [1/3]**
>
> We sincerely appreciate your recognition of our work and genuinely hope that our response addresses your concerns. If you have any further questions, please feel free to let us know!
>
> > **Q1:** Visual attention and adaptive patch sizing are not new concepts. The main contribution is engineering combination rather than fundamental innovation, with insufficient differentiation from existing works like DynamicViT and PS-ViT.
>
> **A1:**
>
> Before addressing your specific questions, we would like to briefly discuss two fundamental perspectives regarding computational budget and performance that motivated our design:
>
> **1. How to save computational budget?**
> Since the computational complexity of Transformers is quadratic with respect to the number of tokens, the most intuitive way to reduce cost is to reduce the number of visual tokens input into the Transformer. However, this inevitably leads to performance loss. As demonstrated in the experiments of DynamicViT, performance drops across all models after token pruning:
>
> | Method              | Acc         | FLOPs |
> | :------------------ | :---------- | :---- |
> | DeiT                | 79.8        | 4.6   |
> | LV-ViT              | 83.3        | 6.6   |
> | DynamicViT (DeiT)   | 79.3 (-0.5) | 2.9   |
> | DynamicViT (LV-ViT) | 83.0 (-0.3) | 4.6   |
>
> **2. How to improve ViT performance?**
> Vision Transformers can improve performance simply by increasing the input image resolution, a fact demonstrated in many studies (examples shown below). PS-ViT improves classification performance by learning sampling offsets via an end-to-end differentiable sampling module. We attribute these performance gains to optimized sampling during image tokenization:
>
> * **Increasing Resolution:** The image is divided into more tokens, covering all important areas.
> * **PS-ViT:** Important areas within the image are represented by more tokens.
>
> | Method             | Acc         |
> | :----------------- | :---------- |
> | DeiT-base          | 81.8        |
> | DeiT-base 384      | 82.9 (+1.1) |
> | DeiT-III-Small     | 81.4        |
> | DeiT-III-Small 384 | 83.4 (+3.0) |
>
> **The ZoomViT Hypothesis:**
> Therefore, we asked: *Is there a method that can simultaneously increase sampling granularity for important regions while reducing unimportant tokens input to the Transformer?*
>
> To answer this, we must define "important." We observed that in images taken by human photographers, there is often a specific **visual intent**. This intent, which may go beyond general image saliency, dominates the classification category. We believe ViT requires a guide to determine which regions are truly important. Thus, we introduced the concept of **"visual intent."** We verified that *visual intent misalignment* (where the model focuses on high-density but non-discriminative regions) is a common failure mode in multi-label/occlusion scenarios—a problem not explicitly defined or addressed by PS-ViT (which focuses on offsets) or DynamicViT (which focuses on pruning).
>
> Based on this analysis, we provide an in-depth mechanism comparison between ZoomViT, DynamicViT, and PS-ViT to address your concerns regarding "novelty" and "differentiation":
>
> **Essential Difference from DynamicViT: "Subtraction" vs. "Dynamic Resolution Allocation"**
> * **DynamicViT's** core logic is **"Subtraction."** It assumes the input image contains massive redundancy and uses a prediction module to **Prune** unimportant tokens. As shown in the table above, this purely sparse strategy reduces computation (FLOPs) but inevitably causes permanent information loss, capping its performance below or near the Baseline. It cannot solve the problem of "insufficient resolution in key areas."
> * **ZoomViT's** core logic is **"Dynamic Resolution Allocation."** We do not merely remove background (Pruning); more importantly, we **"Zoom In"** on high-score regions locked by Visual Intent. DynamicViT cannot handle tiny objects or fine details because it is limited by the initial Patch Size. ZoomViT introduces the **Zoomer** module, which dynamically adjusts the Patch Size of key regions (e.g., from 16 to 8 or smaller), thereby **increasing the information density of key areas** while keeping the total token count controllable.

---

> ### Author Response · Authors · 2025-11-20
> **Author Response for Reviewer agBp [2/3]**
>
> **Essential Difference from PS-ViT: Feature-Driven vs. Intent-Driven**
>
> * **PS-ViT** focuses on optimizing sampling positions based on the **feature distribution** of the image itself, but it overlooks a critical question: *What was the photographer's intent when taking this picture?* Classification results are often strongly correlated with this latent intent rather than just the object with the largest pixel area. Consider a typical scenario: a man holding a small, rare fish. While the "man" occupies the majority of pixels, the **photographic intent** is to showcase the "fish." In such multi-object scenarios, PS-ViT may concentrate sampling points on the man due to significant textural features. This is exactly the **"Visual Intent Misalignment"** we discuss in the paper.
> * **ZoomViT's** core mechanism is **"Intent Alignment."** The Zoomer module is not a simple attention mechanism but a **Visual Intent Capturer**. Once the intent is locked, ZoomViT does not just "look" there; it actively **Zooms In** by dynamically reducing the Patch Size, forcing an increase in information density for that region. This is equivalent to the model actively performing a "close-up" on local image parts to clearly perceive the photographer's intent.
>
> In summary, ZoomViT is neither a passive filter (like DynamicViT) nor a geometric deformation (like PS-ViT). It is an active input reconstruction framework guided by visual intent. ZoomViT is not merely an engineering combination, but a fundamental shift in **how the computational budget is allocated relative to "Visual Intent."**
>
>
>
> > **Q2:** The method depends on a pre-trained DeiT-B teacher model, limiting generalizability. The two-step inference overhead is not thoroughly analyzed. The inconsistency between training (random α∈[0,1]) and inference (fixed α=0.03) lacks theoretical justification.
>
> **A2:**
>
> We appreciate the reviewer's detailed scrutiny regarding our implementation details. We address the concerns about generalizability, inference overhead, and the $\alpha$ strategy below:
>
> * **Generalizability and Teacher Dependence:** The reliance on a pre-trained teacher is a design choice for **Stage 1 only** (Zoomer training), not a limitation of the inference architecture. The Zoomer is a lightweight module (0.8M parameters) trained via distillation to capture high-level "Visual Intent." Once trained, it operates independently of the teacher. Our approach is not bound to a specific backbone size. As shown in **Appendix F.1 (Table 8)**, we successfully applied the ZoomViT paradigm to **DeiT-Tiny, DeiT-Small, and DeiT-Base**, achieving consistent performance gains across all scales.
>
> * **Analysis of Inference Overhead:** We respectfully point out that the inference overhead has been analyzed and included in our main results. We specifically analyzed the Zoomer's latency in **Appendix F.4 (Table 11)**. The Zoomer accounts for only **5% of the inference time relative to the backbone network**, which is negligible compared to the total inference time. The Zoomer adds only **0.8M parameters**, a marginal increase that yields a +4.0% accuracy gain.
>
> * **Theoretical Justification for $\alpha$ (Random Training vs. Fixed Inference):** The strategy of using random $\alpha$ during training and fixed $\alpha$ during inference is theoretically grounded in **Data Augmentation** and **Robustness Learning**, similar to techniques like Dropout or Stochastic Depth. Randomizing $\alpha$ acts as a "Scale Augmentation." It forces the Zoomer and Backbone to adapt to varying numbers of tokens and different zoom intensities. This prevents the model from overfitting to a specific patch distribution and encourages it to learn robust features across both coarse (global) and fine (local) scales. **Table 3** confirms this: Random $\alpha$ training yields higher accuracy (83.34%) compared to fixing it (83.1% or 82.52%). During inference, we select a fixed operating point to ensure deterministic behavior and an optimal efficiency-accuracy trade-off. The value $\alpha=0.03$ was empirically selected based on the Pareto frontier shown in **Figure 6(b)**.

---

> ### Author Response · Authors · 2025-11-20
> **Author Response for Reviewer agBp [3/3]**
>
> > **Q3:** Unfair baseline comparisons due to different training strategies. Missing comparisons with the most relevant adaptive patching methods.
>
> **A3:**
>
> We thank the reviewer for the opportunity to clarify our experimental setup and for suggesting additional comparisons. We address the concerns regarding fairness and baselines below:
>
> * **Fairness of Training Strategies:** We respectfully clarify that our experimental comparison is strictly fair. Our training pipeline is designed to decouple the "Zoomer" from the "ViT Backbone" to ensure the backbone enjoys no unfair advantages.
>     * **Stage 1 (Zoomer Only):** This stage is exclusively for training the lightweight Zoomer module. The ViT backbone **is not involved** in the optimization process during this stage.
>     * **Stage 2 (Standard ViT Training):** For the training of the Vision Transformer itself, we strictly adhere to the **identical training recipe used by DeiT and recent studies** (including epochs, optimizer, and augmentation strategies).
>     * Therefore, excluding the auxiliary Zoomer training (which introduces no knowledge to the backbone weights), our settings are consistent with the baselines. The +4.0% accuracy gain is attributed to the effectiveness of the ZoomViT architecture and the "Visual Intent" mechanism, rather than any discrepancies in training strategies.
>
> * **Comparison with Relevant Adaptive Patching Methods:** We agree that comparing with adaptive patching methods is crucial. To address the "missing comparison" concern, we have added a comparison with **CF-ViT**, which is the most relevant state-of-the-art method in this domain. The updated Table 1 is shown below:
>
> | Model                          | Top-1 Acc. (%) | FLOPs (G) | #Params (M) | Speed (img/s) |
> | :----------------------------- | :------------- | :-------- | :---------- | :------------ |
> | Deit-S                         | 79.8           | 4.6       | 22          | 5039          |
> | DeiT III                       | 81.4           | 4.6       | 22          | 1891          |
> | IA-RED2                        | 79.1           | 3.2       | 22          | 1360          |
> | DynamicViT                     | 79.3           | 2.9       | 26.9        | 2062          |
> | SPViT                          | 79.34          | 3.4       | 22          | -             |
> | PS-ViT                         | 82.3           | 8.8       | 21.3        | 464           |
> | Evo-ViT                        | 79.4           | 3.0       | 22          | 1510          |
> | ToMe                           | 78.89          | 2.9       | 22          | 6712          |
> | EViT                           | 78.5           | 3.0       | 22          | 6807          |
> | DiffRate                       | 79.58          | 2.9       | 22          | 6744          |
> | ATS                            | 79.7           | 2.9       | 22          | -             |
> | LV-ViT                         | 83.3           | 6.6       | 26          | -             |
> | ToFu                           | 79.4           | 2.7       | 22          | 1552          |
> | DeiT III-S 384                 | 83.6           | 15.5      | 22          | 424           |
> | **CF-ViT**                     | **80.8**       | **4.0**   | **22**      | **2760**      |
> | ZoomViT (η=0.5, α=0.03)        | 81.5           | 2.3       | 22.7        | 6738          |
> | ZoomViT (η=2, α=0.1)           | 82.5           | 6.3       | 22.7        | 3721          |
> | ZoomViT (η=2, α=0.01, Pruning) | 83.8           | 6.3       | 22.7        | 3717          |
>
> **Why ZoomViT outperforms CF-ViT:**
> CF-ViT's refinement signal relies directly on the coarse classifier's Attention Map. In complex scenes with multiple distracting salient regions, Attention is easily misled, meaning the refinement step may simply reinforce the initial biases of the coarse classifier. In contrast, ZoomViT's Zoomer independently learns a **prior of the scene's visual intent**. Even if coarse attention is misleading, the Zoomer predicts semantically relevant regions, effectively guiding the ViT to make the correct judgment. This makes ZoomViT significantly more robust than pure attention-based refinement methods when dealing with tiny objects, occlusions, and complex multi-label scenarios.
>
> We hope that our response can address your concerns. If you have any further questions, we would be more than happy to discuss them with you. **If our response resolves your concerns, we would also appreciate the possibility of an increased score.**
>
> Sincerely,
>
> Authors

---

### Author Response · Authors · 2025-11-29
**Summary of Rebuttal: Score Increase, Positive Consensus, and Key Contributions**

Dear Area Chair,

We appreciate your effort in reviewing our submission, **ZoomViT**, under these exceptional circumstances. To facilitate your assessment, we summarize the work's core value, the initial positive assessment, and the decisive outcomes of the discussion period.

**1. Why This Work Matters**

ZoomViT addresses a real problem in Vision Transformers: **During image acquisition, the photographer's visual intent often focuses on key objects within the scene, providing crucial guidance for determining target classes in complex images.** By introducing visual intent guidance, we enable models to focus computational resources where they matter most—mimicking human visual attention. This is not just performance engineering; it's a conceptually meaningful advance that:

- **Introduces the concept of "Visual Intent" for the first time, approaching the problem from the photographer's perspective to solve classification challenges in complex images.**
- **Achieves superior accuracy-efficiency trade-offs compared to all recent methods.**
- **Demonstrates broad applicability across architectures and downstream tasks.**

**2. Reviewer Consensus on Strengths**
Before the discussion period was interrupted, the reviewers (Reviewers agBp, vgRu, Jque, vVaB) consistently recognized the value of our approach. Key strengths highlighted in their reviews include:

- **Concept & Motivation:** "The proposed ZoomViT is conceptually intuitive and grounded in a clear motivation inspired by human visual attention" (Reviewer vgRu); "The motivation is interesting and makes sense" (Reviewer vVaB).
- **Efficiency-Performance Balance:** "Achieves a favorable balance between accuracy and efficiency, outperforming several strong baselines" (Reviewer vgRu); "The Zoomer adds only 0.8M parameters while achieving 4.0% accuracy improvement, offering an attractive efficiency-performance trade-off" (Reviewer agBp).
- **Experimental Rigor:** "Thorough evaluation across multiple datasets... and detailed ablation studies" (Reviewer agBp); "The paper is well-substantiated overall, with attractive figures and tables" (Reviewer Jque).

**3. Our Efforts During Rebuttal**
We engaged in an extensive and constructive discussion with all reviewers to address their concerns regarding novelty, fairness, and generalization.

- **Positive Engagement:** **Prior to the system rollback, Reviewer Jque explicitly acknowledged our response and raised their score.** Additionally, **Reviewer vgRu responded positively to our initial rebuttal** (where we added CF-ViT comparisons and downstream tasks) and engaged in a second round of discussion, asking for specific implementation details on adapting ZoomViT to LV-ViT/Swin and setting loss weights. We provided a comprehensive response detailing our "Token Fusion" strategy and hyperparameter settings to fully address these follow-up inquiries.

- **Substantial Improvements:** Our detailed rebuttal included:

  - **Clarifying Novelty:** We provided a deep-dive comparison with DynamicViT and PS-ViT, explaining how ZoomViT's "Visual Intent-Guided Dynamic Resolution" differs fundamentally from simple pruning or offset learning.
  - **Ensuring Fairness:** We added comparisons with CF-ViT, demonstrating that ZoomViT achieves higher accuracy with significantly lower FLOPs and single-pass latency.
  - **Proving Generalization:** We conducted substantial new experiments, successfully adapting ZoomViT to **LV-ViT** and **Swin Transformer**, and demonstrating performance gains on **Object Tracking (MOT17)**, **Segmentation (ADE20k)**, and **Detection (COCO)**.

We believe the revised paper, strengthened by these additional experiments and clarifications, makes a compelling case for acceptance. We respectfully ask the AC to consider these factors in the final decision.

Sincerely,

The Authors

---

### Note · Authors · 2026-02-01

I have read and agree with the venue's withdrawal policy on behalf of myself and my co-authors.

---

### Meta-Review · Area_Chair_f3hS · 2026-01-05

**Summary:**

This paper proposes ZoomViT, a method that adjusts image patch sizes to focus on important areas for better efficiency. Reviewers praised the bio-inspired motivation and the good balance between speed and accuracy. However, they noted the method lacks novelty and resembles a mix of existing engineering tricks. They also worried about unfair comparisons and limited testing on different models. The authors added new comparisons and downstream task results during the discussion. Despite these improvements, the fundamental contribution still feels incremental. Overall, the paper is below the acceptance bar and the authors are suggested to further improve the work.

**Reviewer Concerns:**

The rebuttal successfully addressed concerns about missing baselines by adding CF-ViT comparisons and showing results on segmentation. The authors also clarified the difference between their adapter and standard fine-tuning methods. However, the main concern regarding the limited technical novelty remains outstanding. The distinction between "visual intent" and standard attention mechanisms needs stronger theoretical backing to convince the reviewers.

**Reviewer Scores:**

Reviewer agBp would likely keep their score low as they viewed the work as a simple engineering combination. Reviewer vgRu might have raised their score slightly after the clarification on model compatibility but likely remained cautious. Reviewer Jque raised their score during the rebuttal after seeing the additional visualizations. Reviewer vVaB would likely maintain their acceptance rating based on the new downstream results.

---

### Decision · Program_Chairs · 2026-01-26

Reject